ecology, evolution

hummingbirds, specialization, diversification

**Author for correspondence:**
Louie M. K. Rombaut
e-mail: lrombaut1@sheffield.ac.uk

# The evolution of the traplining pollinator role in hummingbirds: specialization is not an evolutionary dead end

Louie M. K. Rombaut[1,2], Elliot J. R. Capp[1], Emma C. Hughes[1], Zoë K. Varley[2,3], Andrew P. Beckerman[1], Natalie Cooper[2] and Gavin H. Thomas[1,3]

[1]Department of Animal and Plant Sciences, University of Sheffield, Sheffield S10 2TN, UK
[2]Department of Life Sciences, Natural History Museum London, Cromwell Road, London SW7 5BD, UK
[3]Bird Group, Department of Life Sciences, Natural History Museum Tring, Akeman Street, Tring, Hertfordshire HP23 6AP, UK

(iD) LMKR, 0000-0001-9396-6127; GHT, 0000-0002-1982-6051

Trapliners are pollinators that visit widely dispersed flowers along circuitous foraging routes. The evolution of traplining in hummingbirds is thought to entail morphological specialization through the reciprocal coevolution of longer bills with the long-tubed flowers of widely dispersed plant species. Specialization, such as that exhibited by traplining hummingbirds, is often viewed as both irreversible and an evolutionary dead end. We tested these predictions in a macroevolutionary framework. Specifically, we assessed the relationship between beak morphology and foraging and tested whether transitions to traplining are irreversible and lead to lower rates of diversification as predicted by the hypothesis that specialization is an evolutionary dead end. We find that there have been multiple independent transitions to traplining across the hummingbird phylogeny, but reversals have been rare or incomplete at best. Multiple independent lineages of trapliners have become morphologically specialized, convergently evolving relatively large bills for their body size. Traplining is not an evolutionary dead end however, since trapliners continue to give rise to new traplining species at a rate comparable to non-trapliners.

## 1. Introduction

For plants that are widely dispersed across a landscape, there is a premium in attracting high-fidelity long-range pollinators and excluding low-fidelity short-range pollinators [1–7]. Trapliners are pollinators that visit widely dispersed flowers along circuitous foraging routes. Widely dispersed plant species may gain an advantage from having adaptations in floral morphology, such as nectar spurs or long corolla tubes, that allow access to rich nectar rewards for traplining pollinators while barring access to non-trapliners [8–11]. For specialization as a trapliner to be profitable, plants must offer adequate rewards to compensate for the energetic cost of travelling between widely dispersed flowers and the opportunity cost of ignoring flowers of other species in the same vicinity [12]. Trapliners should in turn evolve morphological adaptations, such as long bills, that allow them to access the nectar of such flowers [13–17]. Through coevolution with the flowers of the various species they pollinate, trapliners may therefore become more morphologically and ecologically specialized than their non-traplining counterparts. This hypothesis on the coevolution between guilds of widely dispersed flowers and traplining pollinators inspires several macroevolutionary predictions, which we test in this study of hummingbirds.

The most basic of these predictions is that the evolution of traplining should entail convergent morphological specialization. Specifically, we addressed the prediction that through coevolution with the flowers they pollinate, trapliners should evolve relatively large bills for their body size and a higher wing surface area relative to body size [18,19]. In evolving morphological specialization to a

subset of flowers, trapliners may experience more rapid rates of morphological evolution than the average non-trapliner due to directional selection driven by reciprocal coevolution between flowers and pollinators [20,21]. In particular, as flowers evolve longer corolla tubes, hummingbirds should quickly evolve longer bills. Directional selection for longer bills may also lead to lineages of trapliners breaking ancestral allometric constraints on bill evolution, so that clades of trapliners have weaker evolutionary correlations between bill and body size.

Ecological specialization on a subset of resources could be hypothesized to be an 'evolutionary dead end' in the sense that evolutionary reversals back to a generalized ecology are rare and specialized species rarely give rise to new specialized species [22,23]. The evolution of an increasingly specialized morphology in hummingbirds to exploit specific sets of flowers may make subsequent reversal to a generalist niche more unlikely. While an adaptive ramp may be available for hummingbirds to become increasingly morphologically specialized through gradual coevolution with flowers, reversals may be hindered by an absence of a gradual adaptive ramp in the reverse direction due to competition with short-billed hummingbird species for short-tubed flowers. An ecologically specialized hummingbird may also be more vulnerable to extinction than a generalist and may have reduced potential to spawn new ecologically distinct species. This effect has been found in some groups of organisms [24,25]. Hence, clades of specialist hummingbirds may have lower rates of diversification than generalist clades. On the other hand, clades of specialist trapliners may be able to diversify in the specific flowers on which they feed, supporting high rates of diversification, the opposite of what the evolutionary dead ends hypothesis would predict. Evidence in the literature for specialization being an evolutionary dead end is currently mixed [26,27]. Here, we test these ideas on hummingbirds using phylogenetic comparative methods to characterize diversification and rates of morphological evolution in relation to evolutionary transitions in foraging ecology.

## 2. Methods

### (a) Data

We took three-dimensional (3D) scans of the entire bill and linear measurements of bill length, bill width, bill depth, wing length and tail length from one male museum specimen for each of 289 species at the Ornithological Collection of the Natural History Museum in Tring (UK). Using data from ref. [28], we estimate that intraspecific variance in body mass is only approximately 1.3% of interspecific variance in our data and intraspecific variance in bill length is only approximately 1.1% of interspecific variance, so intraspecific variation in morphological traits is unlikely to impact our results on this scale of analysis. We 3D-scanned and landmarked bills of specimens as described in ref. [29]. There were four fixed landmarks and three semi-landmark curves of 25 points each. We used the R package geomorph [30] to perform landmark alignments and extract principal components (PCs) of shape variation as well as bill centroid size (mm), an overall measure of bill size defined as the square root of the sum of squared Euclidean distances from the centroid to each of the landmarks. We retained the first three PCs describing greater than 95% of the variation in shape. We obtained data on the mean body mass of each species in grams from ref. [31]. All morphological measurements were $\log_e$ transformed.

We were able to classify 238 hummingbird species (approx. 80% of all genera) as either trapliners (70), territorial (104) or opportunists (64). We also performed sensitivity analyses

where opportunists and territorial species were classed as non-trapliners. We obtained information on the foraging behaviour of hummingbird species from the 'Handbook of the Birds of the World' [32], the Cornell Lab of Ornithology's 'Birds of the World' online database (www.birdsoftheworld.org) [33] and the 'Hummingbirds of North America' [34]. There can be strong sexual dimorphism within hummingbird species in both morphology and foraging behaviour [7,14,35,36]. Since we obtained morphological data for male museum specimens, species were classified according to the foraging behaviour of males wherever sex-specific information was available. There are 12 species in our assembled dataset where foraging behaviour is described for both sexes, and there is sexual dimorphism in six of these. The terms 'traplining' and 'territorial' are regularly used as a dichotomy to describe hummingbird foraging behaviour in the literature. We also considered descriptions such as 'visiting dispersed flowers' or 'following circuitous foraging routes' as further support for classifying a species as a trapliner and descriptions such as 'feeding on clumps of flowers' or 'displaying aggression towards other hummingbirds' as further support for classifying a species as territorial. Species we classified as opportunists are those that are described as 'facultatively territorial' or displaying territorial behaviour seasonally or in some geographical locations but not others. Some hummingbird species are described as 'filchers', sneaking into the territories of other hummingbirds to feed on flowers. Since many of these species are also described as 'facultatively territorial', they were classified as opportunists. We acknowledge the potential for error in foraging classifications. However, the inclusion of species with uncertain classification into the intermediate opportunist category should increase our power to detect differences between species that are confidently classed as trapliners and those confidently classed as territorial.

We used two alternative published phylogenies in our comparative analyses [37,38]. Trees from ref. [37] (available from www.birdtree.org) are based on genetic sequence data plus taxonomic imputation for 299 species, while trees from ref. [38] (available from https://tree.opentreeoflife.org) are based on genetic data only for 291 species. We constructed maximum clade credibility trees from phylogenetic posterior distributions using TreeAnnotator [39]. Taxonomic labels for ecological and morphological data were matched to the two phylogenies with taxonomic synonym information from the Cornell Lab of Ornithology's 'Birds of the World' online database (www.birdsoftheworld.org) [33].

### (i) Does the evolution of traplining entail convergent morphological specialization?

We used random forest classification models [40,41] to test whether differences in foraging behaviour are associated with differences in the dimensions of morphological traits. Random forest models use sets of decision trees to classify items according to multiple variables and have an advantage in accounting for potentially complex multi-dimensional relationships between predictor and response variables. First, we performed a PC analysis (PCA) on all morphological traits combined and used the PCs as predictors of foraging behaviour. We also repeated the analysis using phylogenetic PCA. In a second model, we performed a separate PCA on bill centroid size and body mass alone and used these PCs as predictors, as these traits are expected to be most closely associated with foraging behaviour. PC1 accounted for 68% of the variation and PC2 accounted for the remainder. Because this reduced model had the same predictive power as the full model (see §3), we focused on just these two traits in the rest of our analyses. We tuned forest size and number of variables to consider at each split by trying different values and seeing which maximized classification accuracy.

Following initial tuning of forest size and number of variables to consider at each split, we used fivefold cross-validation to estimate classification accuracy. To generate a null expectation of classification accuracy based on observed phylogenetic similarity among species, we simulated the evolution of traits randomly under the Brownian motion model of evolution 1000 times using the fastBM function of the R package phytools (Revell [42]) on rate-scaled trees inferred using BayesTraits v3 with default settings [43] (http://www.evolution.rdg.ac.uk/; see below) and used these randomly simulated traits as predictors of foraging behaviour. We generated simulated data for bill centroid size and body mass independently. As a complementary analysis, we used phylogenetic generalized least squares (PGLS) in the R package caper [44] to test for differences in the slopes and intercepts of allometric relationships between bill size and body size among trapliners, territorialists and opportunists.

## (ii) Does the evolution of traplining entail higher rates of morphological evolution and weaker evolutionary correlation between bill size and body size?

We used the R package ratematrix [45,46] to test for an association between foraging behaviour and rates of morphological evolution in bill size and body size, as well as differences in the evolutionary correlation between bill size and body size. For each discrete state, ratematrix estimates a variance–covariance matrix for morphological traits under a correlated Brownian motion model of evolution. We supplied ratematrix with 1000 stochastic character mappings of the evolutionary history of foraging behaviour on our phylogenies generated using the make.simmap function of phytools [42]. We set normal priors on the phylogenetic root states of bill size and body size with means and standard deviations equal to those estimated from the data. We set a lognormal prior on the evolutionary variances of both traits with a mean of 0.1 and standard deviation of 1.5. For the correlation between traits, we set a uniform prior. We ran four independent Markov chain Monte Carlo (MCMC) chains in parallel for 20 million generations each with a burnin proportion of 0.25. We repeated analyses using 'trapliner' and 'non-trapliner' classifications.

Determining whether differences in rates of morphological evolution are associated with particular character states can be complicated by background rate variation [47]. We therefore used the MuSSCRat model [47], implemented in RevBayes [48] (https://revbayes.github.io/), as an additional test of our hypothesis. This allows us to test whether character states are associated with different rates of evolution beyond what would be expected from random background variation alone. We note, however, that the MuSSCRat code currently available only allows for overall rates in trait evolution, and not correlations between traits, to vary between examined states. We ran four chains for 100 000 generations each with a burnin of 10 000 generations. All priors and settings were the defaults except for the prior on the number of expected number of transitions which we set to 20 based on a prior judgement of how many transitions in foraging ecology there appear to have been in the tree.

To further identify and visualize how rates of morphological evolution vary across the phylogeny independent of foraging behaviour, we used BayesTraits to fit a variable rates model of correlated evolution between bill size and body size. Two chains were run in parallel for 110 million generations each with a burnin of 10 million generations. All other priors and settings were the defaults.

For all analyses involving MCMC sampling, we visualized traces and posterior distributions using Tracer [49]. We checked that the effective sample size for parameters was greater than 200 and that Gelman and Rubin's R diagnostic among chains was less than 1.05.

## (iii) Is traplining an evolutionary dead end?

We tested two predictions of the hypothesis that traplining is an evolutionary dead end. First, we fitted models of evolutionary transitions in foraging behaviour using the fitDiscrete function in the R package geiger [50]. From fitting a full model in which all transition rates between states were estimated, we found that the transition rates from traplining to territorialism and from opportunism to traplining were very close to zero. We therefore fitted a reduced model in which these transition rates were fixed to zero and compared the full and reduced models using the Akaike information criterion (AIC). We used this to test for irreversibility in transitions.

Second, we tested whether traplining is associated with lower net diversification rates. Whether a discrete state is associated with differences in rates of speciation or extinction across a phylogeny can be complicated by background rate variation [51]. We therefore used the R package SecSSE [52] to jointly estimate transition rates and test for state-dependent diversification. We tested models in which rates of speciation and extinction differ between the three foraging states examined against models in which speciation and extinction rates differ between three hidden states and models with constant background rates. We tested models with one, four and six independent transition rates between states. To avoid the possibility of getting stuck in a local likelihood optimum, we ran five independent repetitions of the likelihood maximization algorithm with different starting points as follows: (1) speciation and extinction rates set to their maximum-likelihood estimates (ML) from a simple birth–death model + transition rates set to 1/5 of speciation rates; (2) ML speciation rates double + ½ transition rates; (3) ½ ML speciation rates + double transition rates; (4) double ML extinction rates + ½ transition rates and (5) ½ ML extinction rates + double transition rates. We set num_cycles = Inf and optimmethod = 'subplex'. We selected the maximum of the five estimated likelihoods to compare models. We compared models using AIC scores. We assumed the sampling fraction of phylogenies to be 0.88 for all examined states based on the proportion of taxonomically recognized species (338) represented in the phylogeny. We repeated analyses using the trapliner versus non-trapliner classification of foraging behaviour.

# 3. Results

## (a) Does the evolution of traplining entail convergent morphological specialization?

The random forest classification model predicted foraging behaviour from PCs of all morphological traits with an accuracy of 61% ($\sigma = 6\%$) or 58% if using phylogenetic PCA. However, PCs of body mass and bill centroid size alone are sufficient as predictors to achieve a classification accuracy of 60% ($\sigma = 5\%$). This can be contrasted with the classification accuracy achieved when the random forest model is applied to data simulated under the assumption that morphology evolves independently of foraging behaviour (classification accuracy: 41%; 95% CI = 34–50%). Opportunists might be expected to be intermediate between trapliners and territorialists, bringing down the overall classification accuracy. As expected, when only considering trapliners and territorialists, the overall classification accuracy increases to 75% ($\sigma = 11\%$). The classification accuracy remains 75% when using phylogenetic PCA.

There is a significant difference in the slope of the allometric relationship between bill size and body size between trapliners, opportunists and territorialists (PGLS: $F_{2,207} = 5.71$, $p = 0.004$). The significance of this result is robust to the exclusion of the

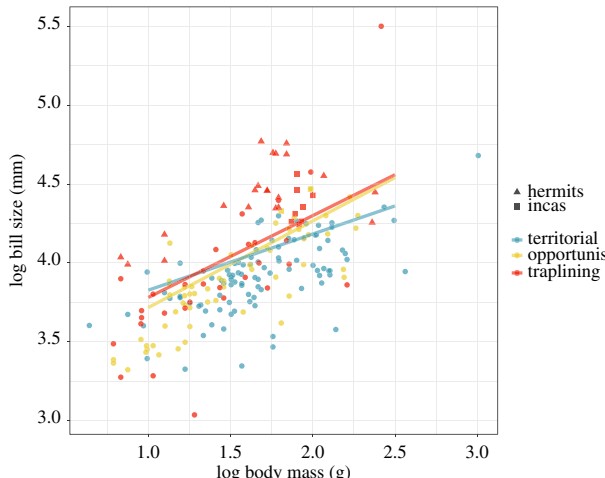

**Figure 1.** Allometry in bill centroid size (mm) and body size (g) of hummingbirds (PGLS): territorialists: log(bill size) = 0.36 × log(body mass) + 3.47; opportunists: log(bill size) = 0.55 × log(body mass) + 3.17; trapliners: log(bill size) = 0.52 × log(body mass) + 3.26. (Online version is in colour.)

evolutionary outlier *Ensifera ensifera*, the sword-billed hummingbird which has an extremely long bill. Between trapliners and opportunists, trapliners tend to have the largest extremes of relative bill size, although the predicted phylogenetic regressions are very close in the two groups. It could be hypothesized that opportunists are therefore ecologically adapted to traplining since they are at least facultatively trapliners. This is not inconsistent with our hypotheses. Trapliners, and opportunists, tend to have relatively larger bills for their body size than territorialists (figure 1). To a large extent, this trend is driven by species of the hermit hummingbird clade (Phaethorninae). Trapliners of the inca clade (*Coeligena*) have independently converged on a hermit-like morphology, as have multiple isolated lineages of trapliners: e.g. *E. ensifera*, *Androdon aequatorialis*, *Doryfera* species, *Polytmus* species and *Myrtis fanny*. Members of the traplining *Chlorostilbon* and *Lophornis* clades, however, fall within the morphological range of non-traplining hummingbirds.

### (b) Does the evolution of traplining entail higher rates of morphological evolution and weaker evolutionary correlation between bill size and body size?

Inferred differences among traplining, territorialist and opportunist lineages in estimated average rates of bill size and body size evolution, and their evolutionary correlation inferred using ratematrix, are sensitive to the phylogeny on which analyses are performed. (The results of all pairwise comparisons can be found in these figures: figure 2; electronic supplementary material, figures S1 and S2.) We used the results of the BayesTraits analyses of variable rates of trait evolution to identify possible causes of the inconsistent results across trees. This showed that the two main differences between the phylogenies are the presence of *Hylonympha macrocerca* and the greater upshift in rates of evolution within the bee hummingbird clade in the Jetz *et al.* [37] phylogeny (electronic supplementary material, figure S3). When we repeat the ratematrix analysis with the evolutionary outlier species *H. macrocerca* pruned from the Jetz *et al.* [37] phylogeny,

there is no longer a significant difference in rates of body size evolution between trapliners, territorialists and opportunists (overlap in Bayesian posterior distributions greater than 5%). However, even when the Jetz *et al.* [37] phylogeny is pruned to have only species that are also present in the McGuire *et al.* [38] phylogeny, there is still a discrepancy between the two sets of analyses. Only when *H. macrocerca* and all members of the bee hummingbird clade are pruned from the Jetz *et al.* [37] phylogeny are the results congruent with analyses on the McGuire *et al.* [38] phylogeny. Given these facts and the significant overlap in posterior distributions for parameters, we conclude that there is no definitive evidence for differences in the average rate of bill size and body size evolution, or the evolutionary correlation between them, for trapliners, territorialists and opportunists. This remains true when considering the binary classification of 'trapliners' and 'non-trapliners' (electronic supplementary material, figure S2). This conclusion is supported by analysis with the MuSSCRat model (electronic supplementary material, figure S4) which indicates that rates of evolution are variable across the hummingbird phylogeny but are unrelated to foraging behaviour.

### (c) Is traplining an evolutionary dead end?

Transitions to an exclusively traplining lifestyle are relatively rare in hummingbirds compared to transitions between territorialism and opportunism (figure 3; electronic supplementary material, figure S5). Reversals from traplining to territorialism are rarer still. We found that a model in which the transition rate from traplining to territorialism was fixed to zero had a better fit than a model in which all transition rate parameters were free to vary (ΔAIC = 4; electronic supplementary material, table S6), suggesting that such transitions are mostly irreversible. Multiple traplining lineages have transitioned to opportunism however, becoming at least facultatively territorial.

Our estimation of ancestral states on the hummingbird phylogeny identifies several non-sister clades of traplining and opportunist species with a likely traplining ancestor: the hermits (Phaethorninae), with approximately 37 species found mostly in the tropical lowlands of South America; the incas (*Coeligena*), an Andean clade with approximately 11 species; the coquettes (*Lophornis*), with approximately 10 species inhabiting South and Central America and emeralds of the genus *Chlorostilbon*, with approximately 18 species found in South and Central America and the Caribbean. Additionally, we identify multiple isolated lineages of trapliners represented by only one or two species.

Based on the results of the SecSSE analysis, there is no evidence to suggest that rates of speciation are generally any different for traplining, territorial or opportunist clades (electronic supplementary material, tables S7–S9). The best-supported models among the models we analysed indicate variation in speciation rates among hidden states unrelated to our foraging classifications (ΔAIC ≫ 4).

## 4. Discussion

Specialization may be expected to result in convergent morphological specialization, elevated rates of morphological evolution and evolutionary dead ends. Our findings challenge these assumptions. Although multiple lineages of trapliners have independently become morphologically specialized to

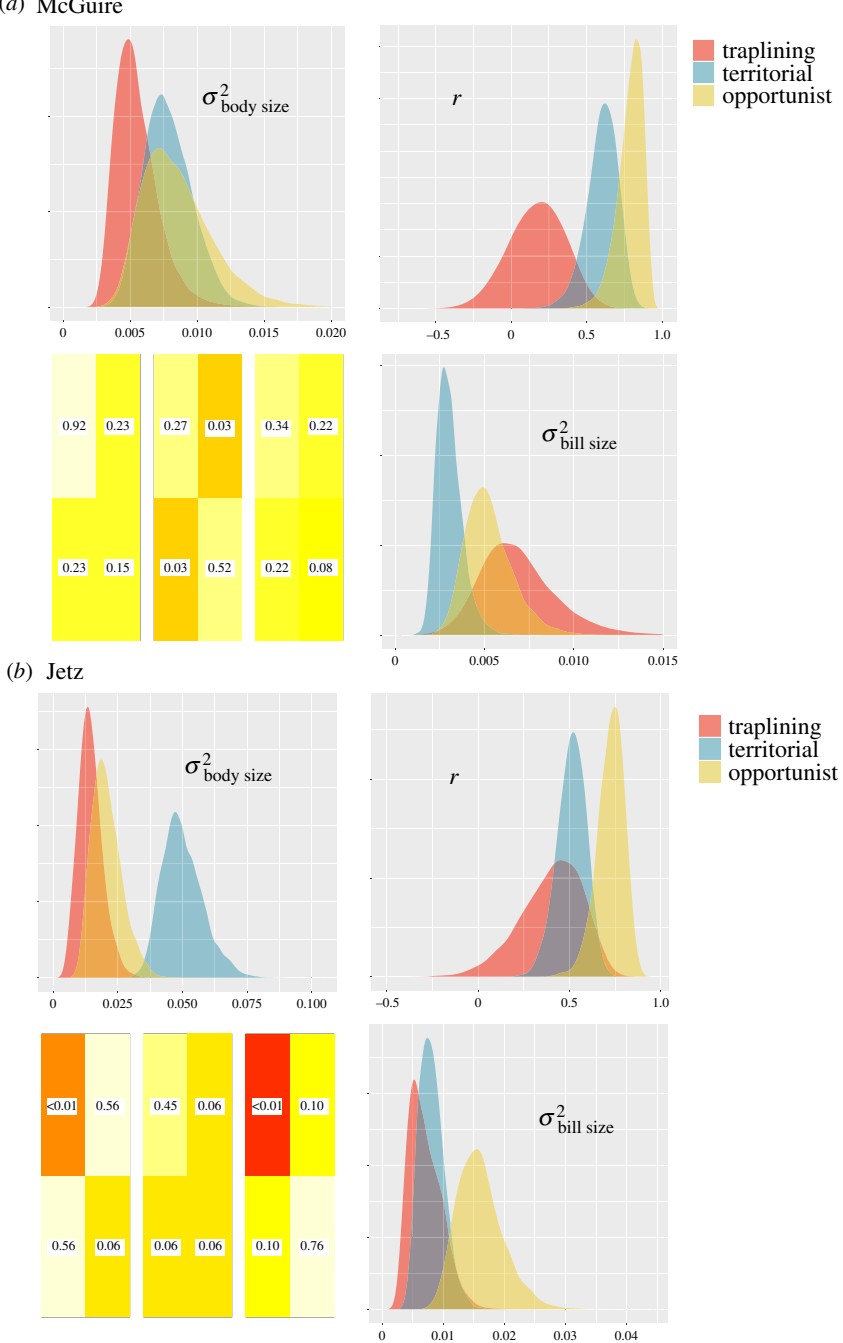

**Figure 2.** Analyses of state-dependent correlated trait evolution with ratematrix. Ratematrix analyses were repeated on the McGuire *et al.* [38] phylogeny (*a*) and the Jetz *et al.* [37] phylogeny (*b*). In each panel, the posterior distributions of evolutionary rate ($\sigma^2$) and correlation (*r*) parameters are compared for different ecological regimes: traplining in red, opportunism in yellow and territorialism in blue. The upper left panel is for rates of body size evolution, the bottom right for rates of bill size evolution and the top right for their evolutionary correlation. The proportion of overlap between posterior distributions in pairwise comparisons between regimes is indicated in the bottom left panel, with darker colours suggesting a more definite difference. Each rectangle in the bottom left panel follows the structure of the $2 \times 2$ evolutionary variance–covariance matrix of bill size and body size evolution. From left to right, the three pairwise comparisons are: opportunist $\times$ territorial, opportunist $\times$ traplining and territorial $\times$ traplining. Numbers within rectangles are proportion overlaps between distributions. (Online version is in colour.)

feed on long-tubed flowers, many other lineages of trapliners remain morphologically unspecialized. While some trapliners such as the sword-billed hummingbird (*E. ensifera*) have experienced greatly elevated rates of morphological evolution, other trapliners have not. Finally, though some clades of trapliners have low rates of diversification, other clades have relatively high rates of diversification for hummingbirds.

Multiple transitions from territorialism to traplining have taken place in the course of hummingbird evolution (figure 3), while reversals from traplining to territorialism are

rare or incomplete in the sense that trapliners only ever become facultatively territorial opportunists. Traplining is essentially a behavioural characteristic of a species' foraging ecology. Behavioural traits are thought to have intrinsically high adaptive plasticity within species and evolutionary lability across species [53]. We infer relatively high rates of transition between lineages that are territorial and lineages that are facultatively territorial opportunists, but once traplining evolves in a lineage, it tends to be conserved. Lineages that evolve to become true trapliners may also quickly become adaptively specialized

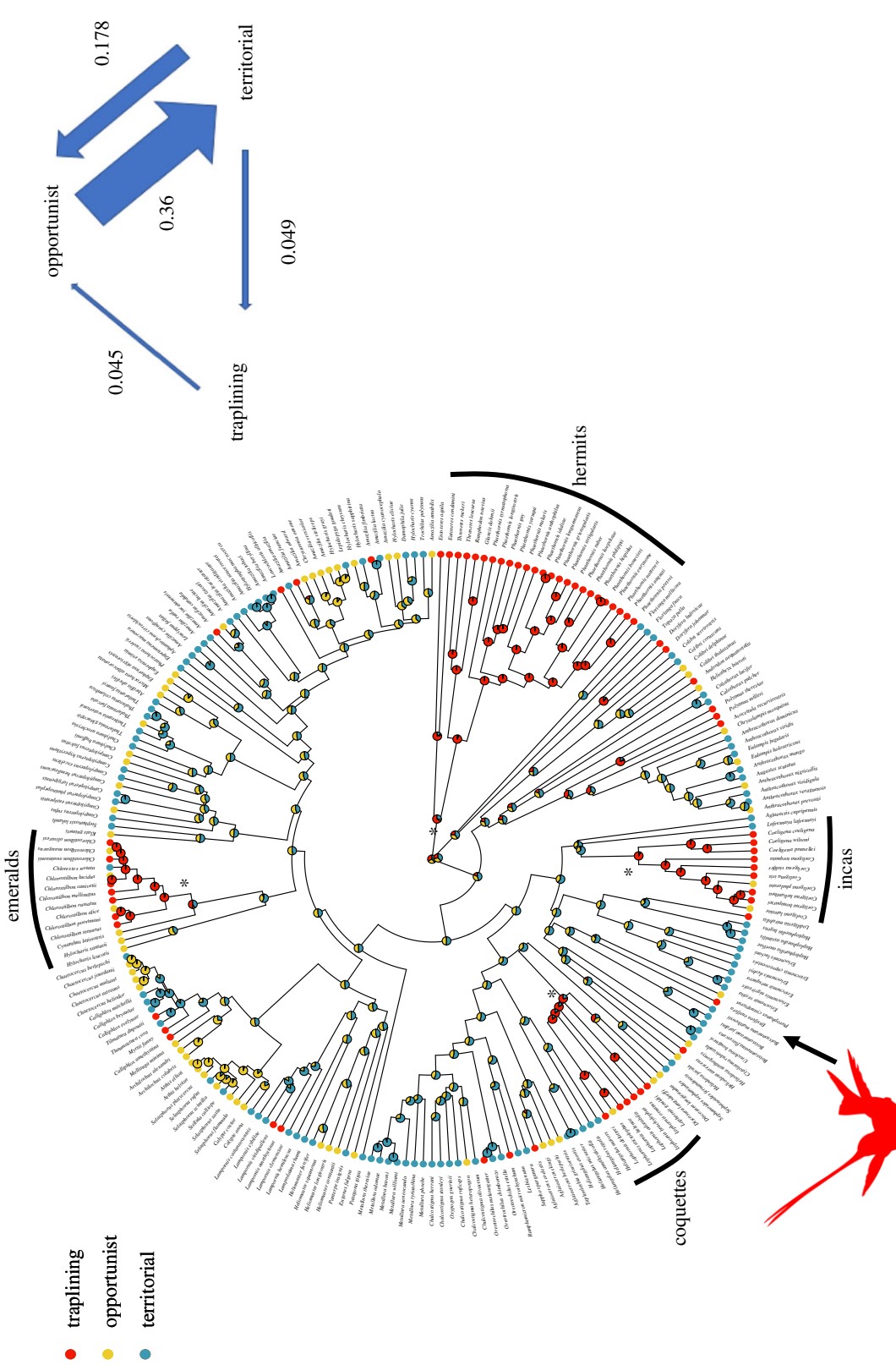

**Figure 3.** The evolution of hummingbird foraging ecology and transition rates between states. The asterisks point to locations on the phylogeny where there was likely an origin of a major traplining clade. Pie charts represent probabilities for ancestral foraging strategies. The red silhouette shows *Ensifera ensifera*. (Online version is in colour.)

to a niche as trapliners. Ecological competition among species, mediated by morphological traits, is an important process in hummingbird community assembly [28,54,55]. It is possible that ecological competition with other hummingbird species may prevent traplining hummingbirds from re-adapting to life as territorialists. Traplining is not an evolutionary dead end however, as in our best-supported model traplining species give rise to new species at a rate comparable to non-trapliners.

Hummingbird diversification is characterized by disparity in rates among major clades [38], but that disparity is unrelated to foraging behaviour. The clade of bee hummingbirds in particular has experienced elevated rates of diversification and stands out as one with a generally elevated rate of evolution in bill size and body size in our analyses. It may be that the ecological niches that bee hummingbirds have occupied as uniquely small hummingbirds have spurred both their diversification and high rates of morphological evolution. We are cautious in this interpretation because elevated rates of trait evolution can arise as an artefact of a combination of trait measurement errors, the effects of short branch lengths and phylogenetic error [51,56,57]. While we have no reason to assume that phylogenetic error is a more severe problem for bee hummingbirds than any other clade, we cannot rule out the possibility that trait measurement error has some effect. This is because bee hummingbirds are among the smallest hummingbirds and even if absolute measurement error is similar among clades, the potential for higher proportional error could disproportionately elevate our measures of trait evolution for this clade.

While we do see evidence for repeated convergent evolution of large bills relative to body size in trapliners, we do not find conclusive evidence for an association between foraging behaviour and rates of bill and body size evolution or the evolutionary correlation between them. One may not necessarily expect to detect elevated rates of trait evolution in sets of living species if adaptive peak shifts which gave rise to the current disparity in morphological traits happened long ago [58]. Adaptive peak shifts may entail punctuational breaks in patterns of trait evolution though equilibrium quickly re-establishes itself. Traplining lineages that have only recently experienced high rates of morphological evolution due to coevolution with flowers may be poorly represented among modern species.

Our macroevolutionary hypotheses necessarily make assumptions about the chain of causation from the spatial distribution of plant species within habitats to the coevolution among the traits of plants and pollinators to the evolution of ecological specialization. Empirical evidence provides support for these assumptions. Widely dispersed plants benefit from being pollinated by traplining hummingbirds [1,4–7], and the bill length of hummingbirds is correlated with the corolla length of the flowers they visit and with ecological specialization [12,17,59,60]. However, there is some evidence to challenge the assumption that traplining hummingbirds are always more ecologically specialized than territorialists. Traplining hermits visit more plant species than other hummingbirds in at least one community [61]. Morphological specialization need not result in ecological specialization if there are minimal trade-offs involved in exploiting a wide range of flowers when resources are abundant, while maintaining adaptations to exploit a narrower range of flowers at times of resource scarcity and high competition [62,63]. Failure of our predictions to hold generally across all clades of trapliners may be a reflection of the fact that these assumptions on the chain of causation are not always met.

In conclusion, we found that the relationship among the evolution of traplining, morphological specialization, rates of morphological evolution and diversification is complex, and it does not lead to simple deterministic outcomes. In a broad comparative study of ecological adaptation, it is difficult to account for complex lineage-specific factors balancing the costs and benefits of evolution towards morphological and ecological specialization. This could be addressed with more detailed field studies on the foraging ecology of different clades of traplining hummingbirds, as well as trapliners in other groups of pollinators.

Data accessibility. Code and data necessary to reproduce the analyses presented in this paper are available from the Dryad Digital Repository: https://doi.org/10.5061/dryad.q83bk3jhp [64].

Authors' contributions. L.M.K.R.: conceptualization, data curation, formal analysis, investigation, methodology, visualization, writing—original draft; E.J.R.C.: data curation; E.C.H.: data curation; Z.K.V.: data curation; A.P.B.: writing—review and editing; N.C.: writing—review and editing; G.H.T.: conceptualization, data curation, funding acquisition, methodology, project administration, resources, supervision.

All authors gave final approval for publication and agreed to be held accountable for the work performed therein.

Competing interests. We declare we have no competing interests.

Funding. L.M.K.R. was funded by a NERC PhD studentship under the Adapting to the Challenges of a Changing Environment (ACCE) DTP. This work was further supported by the European Research Council (grant no. 615709 Project 'ToLERates'), a Royal Society University Research Fellowship (grant no. URF\R\180006), and a NERC Standard grant (grant no. NE/T000139/1).

Acknowledgements. We thank the following people for comments and suggestions on the manuscript: Christopher Cooney, Alexander Slavenko and Thomas Guillerme. We thank M. Adams, H. van Grouw and R. Prys-Jones at the Natural History Museum London and H. McGhie at the Manchester Museum for providing access to and expertise in the ornithological collections and all the volunteer citizen scientists at http://www.markmybird.org for helping us build the beak shape dataset.

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
