## [Peer Review File · Proceedings of the Royal Society B: Biological Sciences]

Review History

RSPB-2021-0934.R0 (Original submission)

Review form: Reviewer 1

Recommendation

Major revision is needed (please make suggestions in comments)

Scientific importance: Is the manuscript an original and important contribution to its field?

Good

General interest: Is the paper of sufficient general interest?

Good

Quality of the paper: Is the overall quality of the paper suitable?

Marginal

Is the length of the paper justified?

Yes

Should the paper be seen by a specialist statistical reviewer?

No

Do you have any concerns about statistical analyses in this paper? If so, please specify them explicitly in your report.

No

It is a condition of publication that authors make their supporting data, code and materials available - either as supplementary material or hosted in an external repository. Please rate, if applicable, the supporting data on the following criteria.

Is it accessible?

Yes

Is it clear?

Yes

Is it adequate?

Yes

Do you have any ethical concerns with this paper?

No

Comments to the Author

Hummingbirds have become quite specialized, yet, there are some species that have evolved further specialization. This interesting study examines a foraging strategy (traplining) which has coevolved with plants, but authors focus only on the bird part. If traplining is an evolutionary dead end, transitions out of that state should be rare and diversification rates for trapliners should be low. Bill morphology seems to be the most important feature for this foraging strategy and authors use different/complementary approaches to describe its evolution. They find that traplining does not match completely the definition of evolutionary dead end as they did find support for transitions out of traplining behaviour being very rare but found no evidence of lower diversification rate for this strategy. The study is interesting, can provide valuable knowledge on foraging evolution in hummingbirds and show evolutionary consequences of specialization.

There are, however, some issues that need work so the research could be properly assessed. My main comment is that the State-Dependent Diversification model is highly overparameterized. It is the right model to use, but the parameter setup needs to be improved. Below, I made some suggestions to tackle this problem so authors can perform the analysis again. These suggestions point to a simpler setup which should be much faster to compute and less likely to find statistical issues. Authors might find different outcome on diversification rates varying across foraging strategies when rerunning this analysis. I also suggest a sensitivity analysis to explore the effect of the foraging strategy classification i.e., repeat the analyses using a different classification. This is necessary to understand the influence of a somehow subjective categorization. I think this should not take a lot of extra work as it is likely authors have built a work pipeline where replacing the input files is not overly complicated. Finally, the manuscript could benefit of including additional details on methods.

Abstract

Line 46. The second and third sentences of the abstract could be rephrased and merged. Also, I think a sentence saying why to be a trapliner requires (or is thought to require) a specialized morphology.

Introduction

Line 68. Maybe it is worth to mention that "widely dispersed flowers" refers to flowers of a single species or a small group of species.

Line 78. Is the bill length the only morphological feature that trapliners evolve? It is worth mentioning briefly the ones you pointed in line 92.

Line 108. I would like to see the beginning of this paragraph slightly modified. The paragraph would read better if the authors mentioned where the concept of evolutionary dead end comes from and what are the criteria (according to classic literature) that an evolutionary dead end should meet to be called as such. If irreversible transitions and lower rates of diversification are the only two points, please make it clearer. So, I would like to see a clear separation between the concept of evolutionary dead end, and the predictions the authors make for traplining.

Line 116. The sentence about specialization increasing the chances of speciation feels out of place in here, it is a bit confusing how to reconcile this with the evolutionary dead end concept which is described above.

Line 118. Please add a couple of sentences explaining what the goals of this study are.

Methods

Line 136. How many landmarks?

Line 152. I understand sex-specific behaviour is not available for all species. For the ones with available information, what percentage of species show difference in sex-specific behaviour? I mean, how rare is that male does something different from the female?

Line 174. I am fully aware of the complications that come with making classifications. I understand that decisions have to be made in order to carry on. What if those facultatively territorial species were assigned to territorial category instead of opportunistic? I think the study will benefit of performing some sensitivity analysis where all analyses are repeated but using facultatively territorials as opportunistic. If general conclusions do not change, then sensitivity analysis could go to supplementary material.

Line 176. To this point, the authors have not described any analysis, so it seems out of place to clarify that analyses were repeated using two different phylogenetic reconstructions. Perhaps is better to describe the tree from reference 35 (and briefly mention that there is another tree around) and in a later stage in methods to say that the analysis was repeated using the one from ref 36 (and can be found in supplementary material).

Line 186. Ecological data means the foraging behaviour?

Line 196. Could you please briefly describe what random forest classification model are? Including the advantages of such a method.

Line 202. It does not seem clear to me how you could perform a PCA using only two variables: bill centroid and body mass.

Line 212. I think more details on the Brownian motion simulation are necessary to fully understand this part of methods.

Line 218. This sentence seems to me to belong to the next section, as suggested by the headline.

Line 252. Could you please clarify what that 20 means and how it was chosen?

Line 254. Should it be morphological evolution instead of just evolution?

Results

Line 328. From this first part of results, I can see that traplining strategy is not only present in hermit hummingbirds but also in other clades e.g., inca clade. However, it is not clear in the text whether inca clade is a sister clade to hermits. I wonder whether authors can provide a more convincing argument about evolutionary convergence of this traplining. From the top of my head I can think of sort of phylogenetic signal tests, but I guess there are more proper and robust ways to test this.

Figure 2. This figure would be easier to read if the panel in yellow is shown along with its axes (pairwise comparison).

Line 388 and Table S6. (Please replace "Fig." by "Table", also in Table S5). In Table S6, I can see there are 38 and 42 free parameters for Constant and dependent model. The state-dependent analysis should not have that many free parameters. Because we have three states, I expect to have three speciation rates (one for each state), plus three extinction rates (but see below) and then the transitions across states. I recommend the authors the following models: one where speciation varies across states but extinction is constant (three lambdas and 1 mu for all states) and one where extinction varies across states but speciation is constant across states (three mus and 1 lambda). And for transitions, it can be convenient to have three models: one with 1 single free rate of transition across the three states (i.e., going back and forth across all states happens at the same rate), one with 6 free transition rates (i.e., shifts across all the three states, back and forth, are possible but those shifts can happen at different rates) and one with similar structure to

model from fitdiscrete (i.e., no return from traplining to territorial nor return from opportunistic to trapling, so 4 rates). Authors can also assume that hidden and examine traits have the same transitions rates so they can use the `diff.conceal = FALSE` in `q_doubletrans` function in `SecSSE` (function that sets the transition matrix). By having a limited number of free parameters (using my recommendation, models will range from 5 to 10 total parameters), the risk of finding only local optima slims. It is important to note that the total number of models to compare will be result of all possible combinations of the options I mentioned above. For instance, I will spell out 4 models: ETD + speciation-dependent + 1 transition rate; ETD + extinction-dependent + 1 transition rate; CTD + speciation-dependent + 1 transition rate; ETD + speciation-dependent + 4 transition rates. All these models are comparable using AIC, as you did in your Table S6.

Discussion

Line 412. I would like to see a more elaborated explanation on “reversals to be incomplete”.

Line 457. This paragraph will read better if authors relate the literature cited in it with the findings of this study, this will provide a more appealing discussion.

Line 479. I think the authors meant “it does not lead”.

Figure 3. The figure will be more informative if there is a visual delimitation of groups (i.e., emeralds, hermits, etc).

Review form: Reviewer 2

Recommendation

Major revision is needed (please make suggestions in comments)

Scientific importance: Is the manuscript an original and important contribution to its field?

Good

General interest: Is the paper of sufficient general interest?

Good

Quality of the paper: Is the overall quality of the paper suitable?

Good

Is the length of the paper justified?

Yes

Should the paper be seen by a specialist statistical reviewer?

No

Do you have any concerns about statistical analyses in this paper? If so, please specify them explicitly in your report.

Yes

It is a condition of publication that authors make their supporting data, code and materials available - either as supplementary material or hosted in an external repository. Please rate, if applicable, the supporting data on the following criteria.

Is it accessible?

Yes

Is it clear?

Yes

Is it adequate?

Yes

Do you have any ethical concerns with this paper?

No

Comments to the Author

In this work the authors ask whether the particular morphological adaptations required from traplining in hummingbirds affects rates of morphological evolution and lineage diversification. They find that transition to traplining feeding is associated with elongation of the bill, but that this does not appear to limit subsequent evolution of those lineages.

The overall premise of the paper is interesting, and I think a suitable suite of analyses have been performed in order to address the question, but I feel there are some pretty significant gaps in how the analyses were performed that require clarification. The authors place the somewhat counter-intuitive (and therefore interesting) results well in the context of the system. The overall clarity of the writing is good, and I am confident the authors can address the highlighted concerns.

Major comments

It appears that the supplementary material was not provided with the submission, preventing evaluation of a large portion of the results. This must be addressed before proceeding.

There must be up-front statements of what the taxonomic sampling is. This is in two places 1) at the start of the data section in the methods, and 2) between the two phylogenetic hypotheses used. Reading the methods, I assumed that taxonomic sampling was equivalent between the phylogenetic hypotheses as there was no mention of sampling; however, in the results it becomes apparent that it is not equal. Please provide raw numbers of species, what was different etc.

There does not appear to be any mention of the ratematrix results the corresponding section (Does the evolution of traplining entail higher rates of evolution and weaker evolutionary correlation between bill size and body size?). The paragraph talks about the BayesTraits analysis, with some mention of correlations, but no specific results. Figure 2 is only BayesTraits results.

I feel there are significant gaps in the paragraph on assessing morphological convergence. First, was a phylogenetic-PCA or non-phylogenetic PCA performed? Given the potential for phylogenetic signal in traits (with largely traplining clades) this step could potentially affect the results in a big way. Then how much variance was explained by each PC axis? Second, what software/program/packages were used for the random forest analysis? How was the initial tuning performed? How were the hyperparameters tuned? There is a lot of information that needs to be clarified here. Third, if a PCA is performed on only two variables (bill centroid size and body mass) is this not just a bivariate regression? It is unclear why a PCA would be needed when two variables are being analyzed.

There needs to be greater exploration of how different models were tested for analyses of character evolution, and subsequently implemented for ancestral state reconstructions. In the methods, the authors outline two models fit using fitDiscrete; first, one where all transition rates are estimated, and one where transitions from traplining to territorialism and from opportunism to traplining were 0. This is a bit worrying because this reads as, after fitting one model, the authors then fit a more extreme version of that model and compared the relative fit. That would not seem to be a meaningful comparison. Are there no other plausible models that could be tested? Second, more models may have been fit, but the supplementary material does not appear to have been provided, meaning table S5 cannot be examined. Also, that the delta AIC was exactly 4 is surprising given the use of 4 as a cutoff value for differentiating models; I would like to know what the non-rounded number is here. Finally, the authors state that transitioning from traplining to territorial is irreversible (line 373), but does occur in their tree (in the

emeralds). In the discussion the authors mention that this transition is incomplete at best, but I think the text could easily be made consistent.

In the results paragraph starting line 361 -- doesn't the analysis performed test for a difference between groups, not pairwise, so there is a significant difference between trapliners+opportunists with territorial, not necessarily that trapliners are greater than both other groups. That is certainly how it appears looking at Figure 1.

Minor comments

Line 69 – it might be worth setting up some background as to why traplining is considered specialization, e.g. bill shape

Line 73: The sentence starting “Widely dispersed...” doesn’t follow logically from previous, suggest editing for clarity. Could maybe even flip sentence ordering with the one before which would take care of the above comment mostly.

I would recommend moving the line 172 – 174 on the numbers of species in each ecological state (possibly with modification) to the start of the corresponding paragraph as knowing how many states there are would make reading the paragraph much easier.

I think there could be more of a setup for the counter argument to the “evolutionary dead-end” hypothesis in the introduction, especially given the results of the study. For example, why can't selection towards a more "generalist" bill shape be equally strong given the costs noted by the authors about becoming a trapliner. It is not quite clear to me why the selection has to be unidirectional – is there a event horizon (for want of a better term) where selection for shorter bills cannot happen?

Line 140: Sentence starting “we obtained data on the mean body mass” I think should be moved to the second sentence of the paragraph

Line 145: nix “and from” and the end of the line

In the data section of the methods I think it would be helpful to state up front how many ecological states there are (3), before providing the description about how species were assigned to those states. It was clearer reading it the second time when I knew what I was expecting.

Line 198: sentence ending “different sets of morphological traits” -> the traits are the same, it's the measurements/landmarks of those traits that differ between ecological states

I really like in the discussion how the authors differentiate between morphological and ecological specialization, this could potentially be highlighted more in the introduction as well.

Review form: Reviewer 3

Recommendation

Major revision is needed (please make suggestions in comments)

Scientific importance: Is the manuscript an original and important contribution to its field?

Good

General interest: Is the paper of sufficient general interest?

Good

Quality of the paper: Is the overall quality of the paper suitable?

Acceptable

Is the length of the paper justified?

Yes

Should the paper be seen by a specialist statistical reviewer?

No

Do you have any concerns about statistical analyses in this paper? If so, please specify them explicitly in your report.

No

It is a condition of publication that authors make their supporting data, code and materials available - either as supplementary material or hosted in an external repository. Please rate, if applicable, the supporting data on the following criteria.

Is it accessible?

Yes

Is it clear?

No

Is it adequate?

Yes

Do you have any ethical concerns with this paper?

No

Comments to the Author

The study provides evidence against the assumption that some specialization traits are “dead-ends” with up-to-date methodology using a substantial dataset. I was a bit skeptical about using a single specimen to represent a whole species, but the intraspecific variation does not seem to be too much -at least much less than the interspecific variation. Overall, the manuscript is well laid out and the writing is easy to follow. The objectives are clearly set, and the findings are adequately summarized and discussed. My two biggest concerns are the categorization of the character states and the presentation of the data. I explain them in more detail below:

State-dependent analyses are very sensitive to the way the trait data is handled. I find it problematic to treat “opportunistic” as a distinct character state. As it was mentioned several times throughout the manuscript, this is an intermediate form of foraging that exhibits both traplining and territorial behaviour with varying degrees. I realize it is difficult to formulate this behaviour as a continuous trait, one option would be to exclude “opportunistic” taxa from the analysis and perform the analysis only with the “traplining” and “territorial” taxa. Alternatively, the presence and absence of territorial (or traplining) behaviour could be coded as 1 and 0 respectively to examine the effect of one state at a time.

Figure 1: The data is not provided to see which data point refers to which species. As it is, there are some interesting outliers that the readers might be interested in knowing more about. OK I see the data file now. It should be cited it in the text, so the readers can easily find it.

Figure 2: The coloured rectangles are extremely difficult to interpret and the figure legend does not help with this either. This part of the figure could be converted to a table instead. Are those p values in the middle of each rectangle? (Same comment for Figure S2)

Figure 3: What does the arrow with a red hummingbird silhouette refer to? The clade boundaries should be more clear; we cannot see which species are included within the emeralds, coquettes...etc.. Most of the species names are not legible on the tree. The legend needs more description for the ancestral reconstruction (e.g., what do the coloured pies represent? What do the asterisks mean?) Some of these might seem obvious, but not for everyone.

Figure S2: Both trees have “outlier” taxa with long branches. Are those the ones that were pruned? Some clarification would be nice. It would also be useful to see the topological differences between the two trees (e.g., by highlighting the differently placed clades).

Figure S5: This should be a table, not a figure (it says table in the main text; same for Figure/Table S6). What constraint is used in the second model? It should be described here so that readers don't have to go back to the main text. In the main text, it says AICc was used, but the table here says AIC. What is n and which one is used?

Thank you for your work and good luck.

Review form: Reviewer 4

Recommendation

Accept with minor revision (please list in comments)

Scientific importance: Is the manuscript an original and important contribution to its field?

Good

General interest: Is the paper of sufficient general interest?

Good

Quality of the paper: Is the overall quality of the paper suitable?

Good

Is the length of the paper justified?

Yes

Should the paper be seen by a specialist statistical reviewer?

No

Do you have any concerns about statistical analyses in this paper? If so, please specify them explicitly in your report.

No

It is a condition of publication that authors make their supporting data, code and materials available - either as supplementary material or hosted in an external repository. Please rate, if applicable, the supporting data on the following criteria.

Is it accessible?

Yes

Is it clear?

Yes

Is it adequate?

Yes

Do you have any ethical concerns with this paper?

No

Comments to the Author

In this study the authors explore how specialization to traplining pollination in hummingbirds impacts the evolution of bill lengths as well as the rates of morphological evolution and diversification. They show that trapliners tend to have longer bills relative to body size compared to other hummingbirds with different foraging behaviors but that rates of morphological evolution and diversification are independent of foraging ecology. They find out that transitions from territorialist to traplining occurred repeatedly throughout the evolutionary history of the hummingbirds without reversal and that trapliners clades did not experience slow down in diversification, refuting the hypothesis that traplining is an evolutionary dead end.

This paper builds on robust comparative methods and has a well written narrative. However because the authors compiled a wealth of different analyses to answer their target questions, it is sometimes difficult to follow through.

Specific comments:

Methods: The first sentence lacks the information about the exact number of species that have been measured. Please also indicate the number of species included in each of the 2 phylogenetic tree. This way one could get a clear picture of the overlap between the morphological, ecological and phylogenetic dataset. Furthermore, Information about sampling in such kind of study is necessary given that the statistical power of PCM is highly dependent on sampling proportion.

L196. A brief explanation of random forest classification model would be useful.

L.212. by rate-scaled trees do you mean trees with branch length proportional to the rate of evolution of the 2 morphological traits? If so, what is the rationale for using such trees when simulating the evolution of these 2 traits?

L.216-218. Could you please specify if you used trait values or PC scores for the pgl's?

L371. Had a better fit rather than was more parsimonious seems more correct

Fig.1 Please mention in the caption that it shows the result of PGLS regressions.

Fig.3. Some of the tips' names are readable but most of them are completely distorted.

Decision letter (RSPB-2021-0934.R0)

11-Jun-2021

Dear Mr Rombaut:

I am writing to inform you that your manuscript RSPB-2021-0934 entitled "The evolution of the traplining pollinator role in hummingbirds: specialisation is not an evolutionary dead end" has, in its current form, been rejected for publication in Proceedings B.

This action has been taken on the advice of referees, who have recommended that substantial revisions are necessary. With this in mind we would be happy to consider a resubmission, provided the comments of the referees are fully addressed. However please note that this is not a provisional acceptance.

Sincerely,
Dr Locke Rowe
mailto:proceedingsb@royalsociety.org

Associate Editor
Board Member: 1
Comments to Author:

Thank you for this interesting and well-written submission. As you will see, the reviewers all found your work to be of interest and significance. However, they also raised substantial concerns regarding the analysis of the data, alongside concerns regarding its presentation. Please address these issues thoroughly in your revision.

Reviewer(s)' Comments to Author:

Referee: 1

Comments to the Author(s)

Hummingbirds have become quite specialized, yet, there are some species that have evolved further specialization. This interesting study examines a foraging strategy (traplining) which has coevolved with plants, but authors focus only on the bird part. If traplining is an evolutionary dead end, transitions out of that state should be rare and diversification rates for trapliners should be low. Bill morphology seems to be the most important feature for this foraging strategy and authors use different/complementary approaches to describe its evolution. They find that traplining does not match completely the definition of evolutionary dead end as they did find support for transitions out of traplining behaviour being very rare but found no evidence of lower diversification rate for this strategy. The study is interesting, can provide valuable knowledge on foraging evolution in hummingbirds and show evolutionary consequences of specialization.

There are, however, some issues that need work so the research could be properly assessed. My main comment is that the State-Dependent Diversification model is highly overparameterized. It is the right model to use, but the parameter setup needs to be improved. Below, I made some suggestions to tackle this problem so authors can perform the analysis again. These suggestions point to a simpler setup which should be much faster to compute and less likely to find statistical issues. Authors might find different outcome on diversification rates varying across foraging strategies when rerunning this analysis. I also suggest a sensitivity analysis to explore the effect of the foraging strategy classification i.e., repeat the analyses using a different classification. This is necessary to understand the influence of a somehow subjective categorization. I think this should not take a lot of extra work as it is likely authors have built a work pipeline where replacing the input files is not overly complicated. Finally, the manuscript could benefit of including additional details on methods.

Abstract

Line 46. The second and third sentences of the abstract could be rephrased and merged. Also, I think a sentence saying why to be a trapliner requires (or is thought to require) a specialized morphology.

Introduction

Line 68. Maybe it is worth to mention that “widely dispersed flowers” refers to flowers of a single species or a small group of species.

Line 78. Is the bill length the only morphological feature that trapliners evolve? It is worth mentioning briefly the ones you pointed in line 92.

Line 108. I would like to see the beginning of this paragraph slightly modified. The paragraph would read better if the authors mentioned where the concept of evolutionary dead end comes from and what are the criteria (according to classic literature) that an evolutionary dead end should meet to be called as such. If irreversible transitions and lower rates of diversification are the only two points, please make it clearer. So, I would like to see a clear separation between the concept of evolutionary dead end, and the predictions the authors make for traplining.

Line 116. The sentence about specialization increasing the chances of speciation feels out of place in here, it is a bit confusing how to reconcile this with the evolutionary dead end concept which is described above.

Line 118. Please add a couple of sentences explaining what the goals of this study are.

Methods

Line 136. How many landmarks?

Line 152. I understand sex-specific behaviour is not available for all species. For the ones with available information, what percentage of species show difference in sex-specific behaviour? I mean, how rare is that male does something different from the female?

Line 174. I am fully aware of the complications that come with making classifications. I understand that decisions have to be made in order to carry on. What if those facultatively territorial species were assigned to territorial category instead of opportunistic? I think the study will benefit of performing some sensitivity analysis where all analyses are repeated but using facultatively territorials as opportunistic. If general conclusions do not change, then sensitivity analysis could go to supplementary material.

Line 176. To this point, the authors have not described any analysis, so it seems out of place to clarify that analyses were repeated using two different phylogenetic reconstructions. Perhaps is better to describe the tree from reference 35 (and briefly mention that there is another tree around) and in a later stage in methods to say that the analysis was repeated using the one from ref 36 (and can be found in supplementary material).

Line 186. Ecological data means the foraging behaviour?

Line 196. Could you please briefly describe what random forest classification model are?

Including the advantages of such a method.

Line 202. It does not seem clear to me how you could perform a PCA using only two variables: bill centroid and body mass.

Line 212. I think more details on the Brownian motion simulation are necessary to fully understand this part of methods.

Line 218. This sentence seems to me to belong to the next section, as suggested by the headline.

Line 252. Could you please clarify what that 20 means and how it was chosen?

Line 254. Should it be morphological evolution instead of just evolution?

Results

Line 328. From this first part of results, I can see that traplining strategy is not only present in hermit hummingbirds but also in other clades e.g., inca clade. However, it is not clear in the text whether inca clade is a sister clade to hermits. I wonder whether authors can provide a more convincing argument about evolutionary convergence of this traplining. From the top of my head I can think of sort of phylogenetic signal tests, but I guess there are more proper and robust ways to test this.

Figure 2. This figure would be easier to read if the panel in yellow is shown along with its axes (pairwise comparison).

Line 388 and Table S6. (Please replace “Fig.” by “Table”, also in Table S5). In Table S6, I can see there are 38 and 42 free parameters for Constant and dependent model. The state-dependent analysis should not have that many free parameters. Because we have three states, I expect to have three speciation rates (one for each state), plus three extinction rates (but see below) and then the transitions across states. I recommend the authors the following models: one where speciation varies across states but extinction is constant (three λ s and 1 μ for all states) and one where extinction varies across states but speciation is constant across states (three μ s and 1 λ). And for transitions, it can be convenient to have three models: one with 1 single free rate of transition across the three states (i.e., going back and forth across all states happens at the same rate), one with 6 free transition rates (i.e., shifts across all the three states, back and forth, are possible but those shifts can happen at different rates) and one with similar structure to model from fitdiscrete (i.e., no return from traplining to territorial nor return from opportunistic to trapling, so 4 rates). Authors can also assume that hidden and examine traits have the same transitions rates so they can use the `diff.conceal = FALSE` in `q_doubletrans` function in `SecSSE` (function that sets the transition matrix). By having a limited number of free parameters (using my recommendation, models will range from 5 to 10 total parameters), the risk of finding only local optima slims. It is important to note that the total number of models to compare will be result of all possible combinations of the options I mentioned above. For instance, I will spell out 4 models: ETD + speciation-dependent + 1 transition rate; ETD + extinction-dependent + 1 transition rate; CTD + speciation-dependent + 1 transition rate; ETD + speciation-dependent + 4 transition rates. All these models are comparable using AIC, as you did in your Table S6.

Discussion

Line 412. I would like to see a more elaborated explanation on “reversals to be incomplete”.

Line 457. This paragraph will read better if authors relate the literature cited in it with the findings of this study, this will provide a more appealing discussion.

Line 479. I think the authors meant “it does not lead”.

Figure 3. The figure will be more informative if there is a visual delimitation of groups (i.e., emeralds, hermits, etc).

Referee: 2

Comments to the Author(s)

In this work the authors ask whether the particular morphological adaptations required from traplining in hummingbirds affects rates of morphological evolution and lineage diversification. They find that transition to traplining feeding is associated with elongation of the bill, but that this does not appear to limit subsequent evolution of those lineages.

The overall premise of the paper is interesting, and I think a suitable suite of analyses have been performed in order to address the question, but I feel there are some pretty significant gaps in how the analyses were performed that require clarification. The authors place the somewhat counter-intuitive (and therefore interesting) results well in the context of the system. The overall clarity of the writing is good, and I am confident the authors can address the highlighted concerns.

Major comments

It appears that the supplementary material was not provided with the submission, preventing evaluation of a large portion of the results. This must be addressed before proceeding.

There must be up-front statements of what the taxonomic sampling is. This is in two places 1) at the start of the data section in the methods, and 2) between the two phylogenetic hypotheses used. Reading the methods, I assumed that taxonomic sampling was equivalent between the phylogenetic hypotheses as there was no mention of sampling; however, in the results it becomes apparent that it is not equal. Please provide raw numbers of species, what was different etc.

There does not appear to be any mention of the ratematrix results the corresponding section (Does the evolution of traplining entail higher rates of evolution and weaker evolutionary correlation between bill size and body size?). The paragraph talks about the BayesTraits analysis, with some mention of correlations, but no specific results. Figure 2 is only BayesTraits results.

I feel there are significant gaps in the paragraph on assessing morphological convergence. First, was a phylogenetic-PCA or non-phylogenetic PCA performed? Given the potential for phylogenetic signal in traits (with largely traplining clades) this step could potentially affect the results in a big way. Then how much variance was explained by each PC axis? Second, what software/program/packages were used for the random forest analysis? How was the initial tuning performed? How were the hyperparameters tuned? There is a lot of information that needs to be clarified here. Third, if a PCA is performed on only two variables (bill centroid size and body mass) is this not just a bivariate regression? It is unclear why a PCA would be needed when two variables are being analyzed.

There needs to be greater exploration of how different models were tested for analyses of character evolution, and subsequently implemented for ancestral state reconstructions. In the methods, the authors outline two models fit using fitDiscrete; first, one where all transition rates are estimated, and one where transitions from traplining to territorialism and from opportunism to traplining were 0. This is a bit worrying because this reads as, after fitting one model, the authors then fit a more extreme version of that model and compared the relative fit. That would not seem to be a meaningful comparison. Are there no other plausible models that could be tested? Second, more models may have been fit, but the supplementary material does not appear to have been provided, meaning table S5 cannot be examined. Also, that the delta AIC was exactly 4 is surprising given the use of 4 as a cutoff value for differentiating models; I would like to know what the non-rounded number is here. Finally, the authors state that transitioning from traplining to territorial is irreversible (line 373), but does occur in their tree (in the emeralds). In the discussion the authors mention that this transition is incomplete at best, but I think the text could easily be made consistent.

In the results paragraph starting line 361 -- doesn't the analysis performed test for a difference between groups, not pairwise, so there is a significant difference between trapliners+opportunists with territorial, not necessarily that trapliners are greater than both other groups. That is certainly how it appears looking at Figure 1.

Minor comments

Line 69 – it might be worth setting up some background as to why traplining is considered specialization, e.g. bill shape

Line 73: The sentence starting “Widely dispersed...” doesn’t follow logically from previous, suggest editing for clarity. Could maybe even flip sentence ordering with the one before which would take care of the above comment mostly.

I would recommend moving the line 172 – 174 on the numbers of species in each ecological state (possibly with modification) to the start of the corresponding paragraph as knowing how many states there are would make reading the paragraph much easier.

I think there could be more of a setup for the counter argument to the “evolutionary dead-end” hypothesis in the introduction, especially given the results of the study. For example, why can't selection towards a more "generalist" bill shape be equally strong given the costs noted by the authors about becoming a trapliner. It is not quite clear to me why the selection has to be unidirectional – is there a event horizon (for want of a better term) where selection for shorter bills cannot happen?

Line 140: Sentence starting “we obtained data on the mean body mass” I think should be moved to the second sentence of the paragraph

Line 145: nix “and from” and the end of the line

In the data section of the methods I think it would be helpful to state up front how many ecological states there are (3), before providing the description about how species were assigned to those states. It was clearer reading it the second time when I knew what I was expecting.

Line 198: sentence ending “different sets of morphological traits” -> the traits are the same, it's the measurements/landmarks of those traits that differ between ecological states

I really like in the discussion how the authors differentiate between morphological and ecological specialization, this could potentially be highlighted more in the introduction as well.

Referee: 3

Comments to the Author(s)

The study provides evidence against the assumption that some specialization traits are “dead-ends” with up-to-date methodology using a substantial dataset. I was a bit skeptical about using a single specimen to represent a whole species, but the intraspecific variation does not seem to be too much -at least much less than the interspecific variation. Overall, the manuscript is well laid out and the writing is easy to follow. The objectives are clearly set, and the findings are adequately summarized and discussed. My two biggest concerns are the categorization of the character states and the presentation of the data. I explain them in more detail below:

State-dependent analyses are very sensitive to the way the trait data is handled. I find it problematic to treat “opportunistic” as a distinct character state. As it was mentioned several times throughout the manuscript, this is an intermediate form of foraging that exhibits both traplining and territorial behaviour with varying degrees. I realize it is difficult to formulate this behaviour as a continuous trait, one option would be to exclude “opportunistic” taxa from the analysis and perform the analysis only with the “traplining” and “territorial” taxa. Alternatively, the presence and absence of territorial (or traplining) behaviour could be coded as 1 and 0 respectively to examine the effect of one state at a time.

Figure 1: The data is not provided to see which data point refers to which species. As it is, there are some interesting outliers that the readers might be interested in knowing more about. OK I see the data file now. It should be cited in the text, so the readers can easily find it.

Figure 2: The coloured rectangles are extremely difficult to interpret and the figure legend does not help with this either. This part of the figure could be converted to a table instead. Are those p values in the middle of each rectangle? (Same comment for Figure S2)

Figure 3: What does the arrow with a red hummingbird silhouette refer to? The clade boundaries should be more clear; we cannot see which species are included within the emeralds, coquettes...etc.. Most of the species names are not legible on the tree. The legend needs more description for the ancestral reconstruction (e.g., what do the coloured pies represent? What do the asterisks mean?) Some of these might seem obvious, but not for everyone.

Figure S2: Both trees have “outlier” taxa with long branches. Are those the ones that were pruned? Some clarification would be nice. It would also be useful to see the topological differences between the two trees (e.g., by highlighting the differently placed clades).

Figure S5: This should be a table, not a figure (it says table in the main text; same for Figure/Table S6). What constraint is used in the second model? It should be described here so that readers don't have to go back to the main text. In the main text, it says AICc was used, but the table here says AIC. What is n and which one is used?

Thank you for your work and good luck.

Referee: 4

Comments to the Author(s)

In this study the authors explore how specialization to traplining pollination in hummingbirds impacts the evolution of bill lengths as well as the rates of morphological evolution and diversification. They show that trapliners tend to have longer bills relative to body size compared to other hummingbirds with different foraging behaviors but that rates of morphological evolution and diversification are independent of foraging ecology. They find out that transitions from territorialist to traplining occurred repeatedly throughout the evolutionary history of the hummingbirds without reversal and that trapliners clades did not experience slow down in diversification, refuting the hypothesis that traplining is an evolutionary dead end.

This paper builds on robust comparative methods and has a well written narrative. However because the authors compiled a wealth of different analyses to answer their target questions, it is sometimes difficult to follow through.

Specific comments:

Methods: The first sentence lacks the information about the exact number of species that have been measured. Please also indicate the number of species included in each of the 2 phylogenetic tree. This way one could get a clear picture of the overlap between the morphological, ecological and phylogenetic dataset. Furthermore, Information about sampling in such kind of study is necessary given that the statistical power of PCM is highly dependent on sampling proportion.

L196. A brief explanation of random forest classification model would be useful.

L.212. by rate-scaled trees do you mean trees with branch length proportional to the rate of evolution of the 2 morphological traits? If so, what is the rationale for using such trees when simulating the evolution of these 2 traits?

L.216-218. Could you please specify if you used trait values or PC scores for the pgl's?

L371. Had a better fit rather than was more parsimonious seems more correct

Fig.1 Please mention in the caption that it shows the result of PGLS regressions.

Fig.3. Some of the tips' names are readable but most of them are completely distorted.

Author's Response to Decision Letter for (RSPB-2021-0934.R0)

See Appendix A.

RSPB-2021-1736.R0

Review form: Reviewer 1

Recommendation

Accept with minor revision (please list in comments)

Scientific importance: Is the manuscript an original and important contribution to its field?

Good

General interest: Is the paper of sufficient general interest?

Good

Quality of the paper: Is the overall quality of the paper suitable?

Good

Is the length of the paper justified?

Yes

Should the paper be seen by a specialist statistical reviewer?

No

Do you have any concerns about statistical analyses in this paper? If so, please specify them explicitly in your report.

Yes

It is a condition of publication that authors make their supporting data, code and materials available - either as supplementary material or hosted in an external repository. Please rate, if applicable, the supporting data on the following criteria.

Is it accessible?

Yes

Is it clear?

Yes

Is it adequate?

Yes

Do you have any ethical concerns with this paper?

No

Comments to the Author

Line 90. I would like to see the traditional sentence that starts with "In here we..." so the reader can go to methods knowing what to expect.

Line 265. Rates of morphological evolution.

Line 286. The SecSSE analysis is now improved. I suggest the tables have a slightly different structure to make visualization easier. I would have one single table (one table McGuire and other for Jetz's tree) so that the AIC comparison will be done across the 25 different models (i.e., without separating into sub-tables a, b and c). The data will inform whether models with 1 rate are preferred than models with 6 rates for instance.

I have found some inconsistencies on Table S8 (but the same issue could be in the other tables) that have to be addressed to make sure the general conclusions remain the same. Models that are a particular case of a general model but richer in parameters should have a better (i.e., higher) or

equal loglikelihood than the general case. So, the model CR_6 (constant rate with 6 transition rates) is a particular case of the more general model CR_1 (constant rate with 1 transition rate). The model CR_6 should have a better loglikelihood (it should have a better fit, equal fit at very least) than the model CR_1. However, this is not the case CR_6 = -1133 and CR_1 = -1121. I think this same happens in ETDs and CTDs.

I suspect that this is caused by finding local optima rather than the global one. This is a common problem when optimizing likelihoods (especially with complex datasets) in different frameworks. There are two things that I strongly recommend to the authors to consider to minimize/solve this problem.

1) I suggest starting the ML search in different points of the parameter space. If authors are following the help and examples in SecSSE R package, they might be using the lambda and mu estimated from a Birth-Death process as starting points. This is a sensible starting point. Giving the inconsistencies presented in the tables, I think it is necessary that the authors repeat the analysis but with using additional starting points. For instance, they can use the double of the estimated lambda (the one estimated with the BD model). If the authors repeat the analysis with another three or four starting points, the chances of finding local optima only will be smaller. Then, you can take the best likelihood across those five starting points and use it to compare across models and prepare the tables. Here, I paste some lines from a paper (val Els et al 2021, Nature Ecol and Evol) where we used a similar thing:

"...we used five different initial parameter sets. The first set of parameters were the estimates of speciation and extinction from a birth-death model fit to the branching times and with transition rates assumed to be a fifth of speciation rate. For the second set, we doubled the speciation rates of the first set, and halved the transition rates. In the third, we halved the speciation rates of the first set and doubled the transition rates. Similarly, the fourth had doubled extinction rates and halved transition rates, and the fifth had halved extinction rates and doubled transition rates compared to the first set. The highest likelihood of the five starting points was taken as the global optimum and used to compare across models. We used AIC weights – thus penalizing the number of free parameters – to select the best models per analysis."

2) I also recommend that the authors use the `secsse_ml` function with the following arguments:

```
num_cycles = Inf
optimmethod = "subplex"
```

This will run as many cycles as necessary to find the best likelihood possible. If after 10 cycles the algorithm does not find the best likelihood, it will give a warning message (it will not run for an infinite number of cycles). This procedure along with the suggestion 1 minimizes the risk of getting stuck in local optima.

Line 327. Could you please re-phrase this sentence?

Line 320. This is an important paragraph in the discussion, but I think it needs some more work.

It starts talking about diversification rates and is later mixed with discussion about trait evolution. What I think is missing is that, they are not integrated and discussed together.

Line 342. Spatial distribution?

Line 341. I think the word "predictions" might not be the right one in here.

Line 341. I like this paragraph but I would like to explicitly read how the findings of the authors are related to the ideas addressed in the paragraph.

I thank the authors and the editor for the opportunity of reviewing this interesting study. Should you have additional questions, feel free to contact me.

Leonel Herrera-Alsina

Review form: Reviewer 2

Recommendation

Major revision is needed (please make suggestions in comments)

Scientific importance: Is the manuscript an original and important contribution to its field?
Good

General interest: Is the paper of sufficient general interest?
Good

Quality of the paper: Is the overall quality of the paper suitable?
Acceptable

Is the length of the paper justified?
Yes

Should the paper be seen by a specialist statistical reviewer?
No

Do you have any concerns about statistical analyses in this paper? If so, please specify them explicitly in your report.
Yes

It is a condition of publication that authors make their supporting data, code and materials available - either as supplementary material or hosted in an external repository. Please rate, if applicable, the supporting data on the following criteria.

Is it accessible?
Yes

Is it clear?
Yes

Is it adequate?
Yes

Do you have any ethical concerns with this paper?
No

Comments to the Author
Please find comments attached. (See Appendix B)

Decision letter (RSPB-2021-1736.R0)

20-Sep-2021

I am writing to inform you that this version of your manuscript RSPB-2021-1736 entitled "The evolution of the traplining pollinator role in hummingbirds: specialisation is not an evolutionary dead end" has, in its current form, been rejected for publication in Proceedings B.

This action has been taken on the advice of referees, who have recommended that substantial revisions are necessary. With this in mind we would consider a resubmission, provided the comments of the referees are fully addressed. However please note that this is not a provisional acceptance.

I would like to emphasize some points. First, we normally do permit multiple rounds of revisions. I am making an exception here. Second, if an ms is resubmitted, you must address all of

the referee's comments. As you will see, one referee and the AE on this manuscript felt that significant issues in the first round of review were not addressed in the resubmission.

In addition to the referee and AE comments, I would add one. Many have lost confidence in estimates of extinction and speciation from extant trees. One place where this has been discussed recently is in Louca and Pennell *Nature* 580:502–505.

Please find below the comments made by the referees, not including confidential reports to the Editor, which I hope you will find useful.

- 1) A 'response to referees' document including details of how you have responded to the comments, and the adjustments you have made.
- 2) A clean copy of the manuscript and one with 'tracked changes' indicating your 'response to referees' comments document.
- 3) Line numbers in your main document.
- 4) Please read our data sharing policies to ensure that you meet our requirements <https://royalsociety.org/journals/authors/author-guidelines/#data>.

Sincerely,
Dr Locke Rowe
mailto: proceedingsb@royalsociety.org

Associate Editor Board Member

Comments to Author:

The manuscript was seen by two of the original reviewers, one of whom points out that minimal attempt was made to address the original comments. In some cases, these comments were minor, and although they should have been explicitly responded to, in-text alterations could have sufficed. However, there is also only a very limited response to some of the major comments, including dismissive explanation of why the authors opted against the reviewer's suggestion of a phylogenetic or non-phylogenetic PCA, and to the reviewer's point regarding bivariate analysis. Although these points may be addressable, it is against the journal's policy to allow protracted rounds of review.

Reviewer(s)' Comments to Author:

Referee: 2

Comments to the Author(s).

Please find comments attached

Referee: 1

Comments to the Author(s).

Line 90. I would like to see the traditional sentence that starts with "In here we..." so the reader can go to methods knowing what to expect.

Line 265. Rates of morphological evolution.

Line 286. The SecSSE analysis is now improved. I suggest the tables have a slightly different structure to make visualization easier. I would have one single table (one table McGuire and other for Jetz's tree) so that the AIC comparison will be done across the 25 different models (i.e., without separating into sub-tables a, b and c). The data will inform whether models with 1 rate are preferred than models with 6 rates for instance.

I have found some inconsistencies on Table S8 (but the same issue could be in the other tables) that have to be addressed to make sure the general conclusions remain the same. Models that are a particular case of a general model but richer in parameters should have a better (i.e., higher) or equal loglikelihood than the general case. So, the model CR_6 (constant rate with 6 transition rates) is a particular case of the more general model CR_1 (constant rate with 1 transition rate).

The model CR_6 should have a better loglikelihood (it should have a better fit, equal fit at very least) than the model CR_1. However, this is not the case CR_6 = -1133 and CR_1 = -1121. I think this same happens in ETDs and CTDs.

I suspect that this is caused by finding local optima rather than the global one. This is a common problem when optimizing likelihoods (especially with complex datasets) in different frameworks. There are two things that I strongly recommend to the authors to consider to minimize/solve this problem.

1) I suggest starting the ML search in different points of the parameter space. If authors are following the help and examples in SecSSE R package, they might be using the lambda and mu estimated from a Birth-Death process as starting points. This is a sensible starting point. Giving the inconsistencies presented in the tables, I think it is necessary that the authors repeat the analysis but with using additional starting points. For instance, they can use the double of the estimated lambda (the one estimated with the BD model). If the authors repeat the analysis with another three or four starting points, the chances of finding local optima only will be smaller. Then, you can take the best likelihood across those five starting points and use it to compare across models and prepare the tables. Here, I paste some lines from a paper (val Els et al 2021, Nature Ecol and Evol) where we used a similar thing:

"...we used five different initial parameter sets. The first set of parameters were the estimates of speciation and extinction from a birth-death model fit to the branching times and with transition rates assumed to be a fifth of speciation rate. For the second set, we doubled the speciation rates of the first set, and halved the transition rates. In the third, we halved the speciation rates of the first set and doubled the transition rates. Similarly, the fourth had doubled extinction rates and halved transition rates, and the fifth had halved extinction rates and doubled transition rates compared to the first set. The highest likelihood of the five starting points was taken as the global optimum and used to compare across models. We used AIC weights – thus penalizing the number of free parameters – to select the best models per analysis."

2) I also recommend that the authors use the `secsse_ml` function with the following arguments:

```
num_cycles = Inf
```

```
optimmethod = "subplex"
```

This will run as many cycles as necessary to find the best likelihood possible. If after 10 cycles the algorithm does not find the best likelihood, it will give a warning message (it will not run for an infinite number of cycles). This procedure along with the suggestion 1 minimizes the risk of getting stuck in local optima.

Line 327. Could you please re-phrase this sentence?

Line 320. This is an important paragraph in the discussion, but I think it needs some more work. It starts talking about diversification rates and is later mixed with discussion about trait evolution. What I think is missing is that, they are not integrated and discussed together.

Line 342. Spatial distribution?

Line 341. I think the word "predictions" might not be the right one in here.

Line 341. I like this paragraph but I would like to explicitly read how the findings of the authors are related to the ideas addressed in the paragraph.

I thank the authors and the editor for the opportunity of reviewing this interesting study. Should you have additional questions, feel free to contact me.

Leonel Herrera-Alsina

Author's Response to Decision Letter for (RSPB-2021-1736.R0)

See Appendix C.

RSPB-2021-2484.R0

Review form: Reviewer 2

Recommendation

Accept with minor revision (please list in comments)

Scientific importance: Is the manuscript an original and important contribution to its field?

Good

General interest: Is the paper of sufficient general interest?

Good

Quality of the paper: Is the overall quality of the paper suitable?

Good

Is the length of the paper justified?

Yes

Should the paper be seen by a specialist statistical reviewer?

No

Do you have any concerns about statistical analyses in this paper? If so, please specify them explicitly in your report.

No

It is a condition of publication that authors make their supporting data, code and materials available - either as supplementary material or hosted in an external repository. Please rate, if applicable, the supporting data on the following criteria.

Is it accessible?

Yes

Is it clear?

Yes

Is it adequate?

Yes

Do you have any ethical concerns with this paper?

No

Comments to the Author

This is my third reading of this manuscript, and I am extremely pleased to see the detail with which the authors have addressed this latest revision. It enabled me to evaluate the strength of their results in the context of pretty common concerns. That some of the key results don't change while accommodating these concerns only serves to strengthen their conclusions, a more

desirable outcome than disregarding issues raised by reviewers. I still think this is a really interesting paper, and I think it will ultimately be a useful contribution to the field. I don't have any major methodological concerns, most of my in-line comments I feel are subjective with the exception of two, which I highlight below.

Minor comments

Line 277: would move the sentence "the results of all pairwise..." inside the parentheses of the supplementary figures

Line 338: Sentence starting "This could be explained" took me a couple of reads, should the sentence be "trapliners also become morphologically specialized to a niche...", perhaps revisit this sentence for clarity.

Decision letter (RSPB-2021-2484.R0)

14-Dec-2021

Dear Mr Rombaut

I am pleased to inform you that your manuscript RSPB-2021-2484 entitled "The evolution of the traplining pollinator role in hummingbirds: specialisation is not an evolutionary dead end" has been accepted for publication in Proceedings B.

The referee has recommended publication, but also suggest some minor revisions to your manuscript. Therefore, I invite you to respond to the referee(s)' comments and revise your manuscript. Because the schedule for publication is very tight, it is a condition of publication that you submit the revised version of your manuscript within 7 days. If you do not think you will be able to meet this date please let us know.

- 1) A text file of the manuscript (doc, txt, rtf or tex), including the references, tables (including captions) and figure captions. Please remove any tracked changes from the text before submission. PDF files are not an accepted format for the "Main Document".
- 2) A separate electronic file of each figure (tiff, EPS or print-quality PDF preferred). The format should be produced directly from original creation package, or original software format. PowerPoint files are not accepted.

3) Electronic supplementary material: this should be contained in a separate file and where possible, all ESM should be combined into a single file. All supplementary materials accompanying an accepted article will be treated as in their final form. They will be published alongside the paper on the journal website and posted on the online figshare repository. Files on figshare will be made available approximately one week before the accompanying article so that the supplementary material can be attributed a unique DOI.

Sincerely,

Dr Locke Rowe

Associate Editor

Comments to Author:

Thank you for the time that you have taken to carefully revise this version of the manuscript. As you will see, the revision has been seen by the reviewer who had previously raised significant concerns, and they are satisfied that the response has improved the manuscript, as am I. Please address the two minor points that they raise. Thank you for this interesting contribution.

Reviewer(s)' Comments to Author:

Referee: 2

Comments to the Author(s).

This is my third reading of this manuscript, and I am extremely pleased to see the detail with which the authors have addressed this latest revision. It enabled me to evaluate the strength of their results in the context of pretty common concerns. That some of the key results don't change while accommodating these concerns only serves to strengthen their conclusions, a more desirable outcome than disregarding issues raised by reviewers. I still think this is a really interesting paper, and I think it will ultimately be a useful contribution to the field. I don't have any major methodological concerns, most of my in-line comments I feel are subjective with the exception of two, which I highlight below.

Minor comments

Line 277: would move the sentence "the results of all pairwise..." inside the parentheses of the supplementary figures

Line 338: Sentence starting "This could be explained" took me a couple of reads, should the sentence be "trapliners also become morphologically specialized to a niche...", perhaps revisit this sentence for clarity.

Author's Response to Decision Letter for (RSPB-2021-2484.R0)

See Appendix D.

Decision letter (RSPB-2021-2484.R1)

16-Dec-2021

Dear Mr Rombaut

I am pleased to inform you that your manuscript entitled "The evolution of the traplining pollinator role in hummingbirds: specialisation is not an evolutionary dead end" has been accepted for publication in Proceedings B.

Your article has been estimated as being 8 pages long. Our Production Office will be able to confirm the exact length at proof stage.

Data Accessibility section

Open Access

Paper charges

Sincerely,

Proceedings B

Appendix A

Reply to Reviewers' Comments

We thank the reviewers of our manuscript for their time and their helpful comments and suggestions. We have taken on board many of their suggestions and we have performed additional analyses.

Multiple reviewers suggested trying a different classification of foraging behaviour to see if our results are sensitive to alternative categorisations. We have tried classifying all opportunists and territorialists together as 'non-trapliners' and repeating our main analyses with this classification, but our conclusions remain qualitatively unaffected. The results can be found in supplementary materials.

Reviewer 1 suggested a specific set of additional analyses to perform with SecSSE. We have redone these analyses as suggested and found that our conclusion remains qualitatively the same.

With this we believe we have addressed the main concerns of the reviewers. Below are replies to specific comments.

Referee 1

Line 68 We have added some clarification that it is usually flowers of several different widely dispersed plant species that trapliners visit, rather than a single plant species.

Line 108-118 This paragraph has been rewritten

Line 136 All details of scanning, landmarking and post-processing for this project are described fully in ref. 27

Line 152 Of the 12 species in our dataset where explicit sex-specific foraging behaviour is available for both males and females, 6 of them have sex-specific foraging differences.

Line 186 We have changed most mentions of foraging ecology to foraging behaviour to avoid confusion

Line 196 We have added an explanation of this in the text.

Line 202. PCA is just a reorientation of axes that define the morphospace. When there are just two traits the first principal component is equivalent to the trend through the data and the second principal component is the residual variation orthogonal to the first principal component.

Line 252 This was chosen based on the size of the tree and our best judgement on how frequent transitions are a priori

Line 328 All of the major traplining clades we mention are not sister to each other. Figure 3 makes it clear where there are likely to be independent origins of traplining.

Referee 2

It is regrettable that the supplementary materials were not available to you. The other reviewers were able to access the supplementary materials. We suggest that you contact the editor for assistance.

A non-phylogenetic PCA was used. We feel that this choice is not likely to significantly affect our results since both methods are simply reorientations of the morphospace axes. When there are just two traits the first principal component is equivalent to the trend through the data and the second principal component captures the residual variation orthogonal to the first principal component. The package used for the PCA has a citation in the references. Initial tuning is a step that is commonly performed initially with a random forest model where you look at several different variable options and see which performs the best. The goal is to set a forest size such that further increases do not greatly improve classification accuracy and with the number of variables considered at each split the goal is to find an optimum.

The reduced transition rate model is a special case of the full model where certain transition rates are fixed at zero. This is conceptually equivalent to testing whether certain transition rates are significantly different from zero, except that it's done in a model simplification framework. In any case, following the advice of another reviewer we have tried different numbers of unique transition rates (2,4,6) in the SecSSE portion of our analyses. In the best-fitting model there are 4 transition rates as we found with fitDiscrete.

Yes, it is true that the F-test is testing for differences among groups with trapliners, territorialists and opportunists. Further to that, figure 1 shows that it is territorialists that are different from the other two as a pair. Between trapliners and opportunists as a pair, it is trapliners that tend to have the largest extremes of relative bill size, although the predicted phylogenetic regressions are very close between the two groups.

Referee 3

We have made some modifications to figures.

Referee 4

We have added number of sampled species for the morphological data and the phylogenies.

Line 196 We have added a description of random forest models

Line 212 Yes, rate scaled trees inferred using BayesTraits have branches scaled to the proportional rate of evolution for each trait. This is not strictly necessary, but the rate

variable model is closer to the true evolutionary process that has produced the present trait distribution.

Line 216 We used trait values.

Fig3. Have you tried opening the file in adobe acrobat or another pdf reader?

Appendix B

In this work the authors ask whether “traplining” is an evolutionary dead-end, since it evolves the evolution of novel bill morphologies that can be inhibitive of feeding from other sources. They conclude it is not a dead-end since the speciation rate for these lineages is comparable to non-traplining species.

This is my second reading of the manuscript having been initially encouraged by the research question and methods performed. However, I’m extremely disheartened to see the brevity of the response to reviewer document (especially to reviewer 3). From the original submission two of my major comments were not addressed in the response document, and none of the inline comments. Given the effort put in by reviewers to provide feedback to improve the manuscript, I strongly advise the authors to re-consider their approach to this document in the future. Addressing in-line comments in such a document is especially minimal effort. Checking my notes from last time I can see that only some of my in-line comments have been addressed. This may be fine, the author may have a different opinion or I may have mis-interpreted something, but this has not been addressed in the response. Overall I am left with the feeling that the authors have tried to patch the document, rather than systematically address the concerns of the reviewers.

Major comments

First, the following comment from my original reading has been largely disregarded:

I feel there are significant gaps in the paragraph on assessing morphological convergence. First, was a phylogenetic-PCA or non-phylogenetic PCA performed? Given the potential for phylogenetic signal in traits (with largely traplining clades) this step could potentially affect the results in a big way. Then how much variance was explained by each PC axis? Second, what software/program/packages were used for the random forest analysis? How was the initial tuning performed? How were the hyperparameters tuned? There is a lot of information that needs to be clarified here. Third, if a PCA is performed on only two variables (bill centroid size and body mass) is this not just a bivariate regression? It is unclear why a PCA would be needed when two variables are being analyzed.

In their response the authors simply state first that “both methods are simply reorientations of the morphospace axes”, which ignores the whole reason why one would do a phylogenetic-PCA (suggested reading Revell, 2009, as a starting point doi.org/10.1111/j.1558-5646.2009.00804.x). In their response the authors also describe what the tuning does for the random forest, but do not provide any details in the manuscript. The final point regarding the bivariate analysis appears to have been ignored completely.

There are now explicit statements about the taxonomic sampling, but there does not appear to be a mention of which species are non-homologous between the phylogenetic data sets, and what their ecological states are. This is potentially important if they are a rare state and/or positioned near the root of the tree.

A sentence needs to be added to the methods describing briefly what the MuSSCRat is/what it does.

Line 146 The morphological traits are the same, it’s the dimensions of those traits that are different. This was not addressed from the previous submission

The ratematrix results are still largely ignored from the results section. The only sentence is “*Inferred differences between traplining, territorialist and opportunist lineages in estimated average*

rates of bill size and body size evolution, and their evolutionary correlation, are sensitive to the 250 phylogeny on which analyses are performed (fig. 2; fig. S1; fig. S2)." With Fig 2 being the ratematrix results, but doesn't describe any of the results. In particular, the lowest correlation between body size and bill size is for trapliners, which is both interesting and counter-intuitive to me not being familiar with the system. The paragraph starting on line 331 "*While we do see evidence for repeated convergent evolution of large bills relative to body size in trapliners*" I think is meant to address this, but to me the discussion points (adaptive peak shift may be old) doesn't really address this question.

In-line comments

Line 69: This prediction has been made before, I would say the difference here is that the authors are testing the hypothesis with a much more comprehensive sampling strategy

Line 84: sentence ending "niche more unlikely" needs supporting citations, there is a rich body of work on this topic

Line 86: This sentence needs supporting citations as to why more specialized species are more likely to go extinct

Appendix C

Reply to Reviewers

We thank the reviewers and editors for their helpful comments and suggestions on our manuscript. We recognise that our previous response to reviewers did not contain sufficient details on how we addressed many of these comments and suggestions and gave the impression that we had not taken the reviews seriously and in the good faith that they had clearly been made. We apologise for our inadequate response. That is why we have appended to this document a revised reply to reviewers for the first round of revisions, as well as full responses to the most recent round. We hope that our responses contain sufficient details to make clear where and how we have made changes to both the previous and new comments. Below we give a brief overview of the major changes to the manuscript and follow this with a point by point response to each comment.

In the last set of reviews the referees identified several major concerns, many of which we addressed with additional analyses in our revised manuscript. The two primary concerns that required additional analyses were the overparameterization of the SecSSE models and the possible sensitivity of the results to alternative classifications of foraging behaviour. The reviewers also made a range of other valuable comments and suggestions. From the reviews of our resubmission, it is clear that some major concerns about the manuscript remained. Firstly, we thank reviewer 1 for spotting the inconsistency in the likelihoods estimated for the different SecSSE models. By following their suggestions for extra analyses, we believe we have remedied this, and our results remain qualitatively consistent with what we have found with previous analyses. Secondly, reviewer 2 has suggested that a phylogenetic PCA would be more appropriate for our analyses on morphological differences between species that have different foraging habits. While we may not agree that in principle our results should be sensitive to the choice of a phylogenetic vs. non-phylogenetic PCA, we thought that this concern would be more convincingly addressed empirically. That is why we repeated the relevant analyses using a phylogenetic PCA. However, we came to qualitatively the same conclusions.

One of the editors has touched upon a deep controversy in the field of comparative studies of diversification at the moment (Louca & Pennell, 2020- Nature). Multiple qualitatively different models of diversification history are fundamentally non-identifiable from phylogenies. This has led some to wonder whether there is any point at all in trying to estimate speciation and extinction rates from phylogenies. We agree that we should always treat these parameter estimates with caution, however it is also clear from subsequent responses to Louca & Pennell (2020) that there is still value in comparative analyses of diversification for hypothesis testing. This is because not all models are equally biologically plausible *a priori* and a hypothesis-driven approach does not necessarily suffer from the nonidentifiability issues highlighted by Louca & Pennell (2020). In particular we draw attention to recent complementary perspectives on the issue.

From Morlon et al., 2020 – biorxiv :

‘The existence of a large number of congruent models therefore poses no direct challenge to the traditional hypothesis-driven research approach. The only possible concern is the question of model selection consistency: if the true model is not in the set of considered models, do we select the correct hypothesis? This question has not been answered one way or the other and would require thorough investigation in future research.’

From Helmstetter et al., 2021- Systematic Biology (<https://academic.oup.com/sysbio/advance-article/doi/10.1093/sysbio/syab083/6382322>):

‘Yet, even if unidentifiability issues remain in such models they may not be relevant to the questions the models were built to answer, for example when the objective is to determine whether one clade has higher diversification rates than another. In cases like these, it is not the precise values of rates that are important but instead the rate difference i.e. whether rates in one group of lineages are higher than another.’

This latter point is particularly pertinent to our analyses of hummingbird diversification rates since our aim is to test whether such rates vary among lineages characterized by different foraging strategies, rather than modelling variation in rates through time from a lineage through time plot, as was the focus of Louca and Pennell (2020).

Referee 1

Line 90. I would like to see the traditional sentence that starts with “In here we...” so the reader can go to methods knowing what to expect.

We have added this sentence to the end of the Introduction (L95): ‘Here we test these ideas on hummingbirds using phylogenetic comparative methods to characterise diversification and rates of morphological evolution in relation to evolutionary transitions in foraging ecology.’

Line 265. Rates of morphological evolution.

We have changed it to ‘rates of morphological evolution’ in several places.

Line 286 (*this is a substantial comment with multiple points so we have broken it down to make the responses easier to follow*). The SecSSE analysis is now improved. I suggest the tables have a slightly different structure to make visualization easier. I would have one single table (one table McGuire and other for Jetz’s tree) so that the AIC comparison will be done across the 25 different models (i.e., without separating into sub-tables a, b and c). The data will inform whether models with 1 rate are preferred than models with 6 rates for instance.

We have changed the format of the tables as suggested (see tables S7-S9).

I have found some inconsistencies on Table S8 (but the same issue could be in the other tables) that have to be addressed to make sure the general conclusions remain the same. Models that are a particular case of a general model but richer in parameters should have a better (i.e., higher) or equal loglikelihood than the general case. So, the model CR_6 (constant rate with 6 transition rates) is a particular case of the more general model CR_1 (constant rate with 1 transition rate). The model CR_6 should have a better loglikelihood (it should have a better fit, equal fit at very least) than the model CR_1. However, this is not the case CR_6 = -1133 and CR_1 = -1121. I think this same happens in ETDs and CTDs.

I suspect that this is caused by finding local optima rather than the global one. This is a common problem when optimizing likelihoods (especially with complex datasets) in different frameworks. There are two things that I strongly recommend to the authors to consider to minimize/solve this problem.

1) I suggest starting the ML search in different points of the parameter space. If authors are following the help and examples in SecSSE R package, they might be using the lambda and mu estimated from a Birth-Death process as starting points. This is a sensible starting point. Giving the inconsistencies presented in the tables, I think it is necessary that the authors repeat the analysis but with using additional starting points. For instance, they can use the double of the estimated lambda (the one estimated with the BD model). If the authors repeat the analysis with another three or four starting points, the chances of finding local optima only will be smaller. Then, you can take the best likelihood across those five starting points and use it to compare across models and prepare the tables. Here, I paste some lines from a paper (val Els et al 2021, Nature Ecol and Evol) where we used a similar thing:

“...we used five different initial parameter sets. The first set of parameters were the estimates of speciation and extinction from a birth–death model fit to the branching times and with transition rates assumed to be a fifth of speciation rate. For the second set, we doubled the speciation rates of the first set, and halved the transition rates. In the third, we halved the speciation rates of the first set and doubled the transition rates. Similarly, the fourth had doubled extinction rates and halved transition

rates, and the fifth had halved extinction rates and doubled transition rates compared to the first set. The highest likelihood of the five starting points was taken as the global optimum and used to compare across models. We used AIC weights—thus penalizing the number of free parameters—to select the best models per analysis.”

2) I also recommend that the authors use the `secsse_ml` function with the following arguments:

```
num_cycles = Inf
```

```
optimmethod = "subplex"
```

This will run as many cycles as necessary to find the best likelihood possible. If after 10 cycles the algorithm does not find the best likelihood, it will give a warning message (it will not run for an infinite number of cycles). This procedure along with the suggestion 1 minimizes the risk of getting stuck in local optima.

We thank the reviewer for spotting these important issues with our SecSSE analyses. We have now performed the additional analyses suggested. We have 1) started the optimality searches at the five different starting points suggested by the reviewer and 2) we set num_cycles=Inf and optimmethod='subplex'. The results have been incorporated into the manuscript (see tables S7-S9). Importantly, the model likelihoods no longer appear to be stuck on local optima. While specific parameter estimates differ slightly from those previously reported, our biological inferences remain unchanged.

Line 327. Could you please re-phrase this sentence?

We agree that this sentence was not sufficiently clear. We have rephrased and further expanded to more fully explain why an association between rates of trait evolution and rates of speciation should be treated with caution (paragraph starting line 343).

'Hummingbird diversification is characterised by disparity in rates among major clades (38) but that disparity is unrelated to foraging behaviour. The clade of bee hummingbirds in particular has experienced elevated rates of diversification and stands out as one with a generally elevated rate of evolution in bill size and body size in our analyses. It may be that the ecological niches that bee hummingbirds have occupied as uniquely small hummingbirds have spurred both their diversification and high rates of morphological evolution. We are cautious in this interpretation because elevated rates of trait evolution can arise as an artefact of a combination of trait measurement errors, the effects of short branch lengths, and phylogenetic error (51,56,57). While we have no reason to assume that phylogenetic error is a more severe problem for bee hummingbirds than any other clade, we cannot rule out the possibility that trait measurement error has some effect. This is because bee hummingbirds are among the smallest hummingbirds and even if absolute measurement error is similar among clades, the potential for higher proportional error could disproportionately elevate our measures of trait evolution for this clade.'

Line 320. This is an important paragraph in the discussion, but I think it needs some more work. It starts talking about diversification rates and is later mixed with discussion about trait evolution. What I think is missing is that they are not integrated and discussed together.

Line 328 We have split this paragraph so that now there is one paragraph on diversification and one on trait evolution.

Line 342. Spatial distribution?

Line 368: changed to 'spatial distribution'

Line 341. I think the word “predictions” might not be the right one in here.

Line 367: Changed 'predictions' to 'hypotheses'

Line 341. I like this paragraph but I would like to explicitly read how the findings of the authors are related to the ideas addressed in the paragraph.

Line 379: We have added the following sentence to this paragraph: ‘Failure of our predictions to hold generally across all clades of trapliners may be a reflection of the fact that these assumptions on the chain of causation are not always met.’

Referee 2

I feel there are significant gaps in the paragraph on assessing morphological convergence. First, was a phylogenetic-PCA or non-phylogenetic PCA performed? Given the potential for phylogenetic signal in traits (with largely traplining clades) this step could potentially affect the results in a big way. Then how much variance was explained by each PC axis? Second, what software/program/packages were used for the random forest analysis? How was the initial tuning performed? How were the hyperparameters tuned? There is a lot of information that needs to be clarified here. Third, if a PCA is performed on only two variables (bill centroid size and body mass) is this not just a bivariate regression? It is unclear why a PCA would be needed when two variables are being analyzed.

In their response the authors simply state first that “both methods are simply reorientations of the morphospace axes”, which ignores the whole reason why one would do a phylogenetic-PCA (suggested reading Revell, 2009, as a starting point doi.org/10.1111/j.1558-5646.2009.00804.x). In their response the authors also describe what the tuning does for the random forest, but do not provide any details in the manuscript. The final point regarding the bivariate analysis appears to have been ignored completely.

We accept that our previous response was not at all sufficient in addressing these concerns. Below we provide more detail in response to each of the above questions:

- *In our initial submission we used non-phylogenetic PCA and thank the reviewer for reminding us of Revell, 2009. We note that Revell (2009) addresses the issue of controlling for phylogeny when using PCA for dimension reduction (specifically in the case of size-correction) which we do not do. While we believe that our use of non-phylogenetic PCA is justified and appropriate, to address the concern that phylogenetic PCA may be more appropriate than the traditional PCA we used, we repeated the relevant analyses using phylogenetic PCA and found comparable results:*
 - *Line 244: ‘The random forest classification model predicted foraging behaviour from principal components of all morphological traits with an accuracy of 61% ($\sigma=6\%$), or 58% if using phylogenetic PCA.’*
 - *Line 251: ‘As expected, when only considering trapliners and territorialists the overall classification accuracy increases to 75% ($\sigma=11\%$). The classification accuracy remains 75% when using phylogenetic PCA.’*
- *We now include details of the variance explained in the PCA (line 160): ‘PC1 accounted for 68% of the variation and PC2 accounted for the remainder.’*
- *The software package for the random forests is the randomForest library in R (cited as ref. 38).*
- *We have now added an explanation of how the model tuning was performed to the text (line 163): ‘We tuned forest size and number of variables to consider at each split by trying different values and seeing which maximised classification accuracy.’*

- *In answer to the question of whether a PCA on two variables is just a bivariate regression, yes it would yield similar results. Bivariate regression decomposes variation into a trend plus residual variation while a PCA decomposes variation into a trend plus a component orthogonal to the trend. Either approach could have been appropriate, but we chose to use the latter to be consistent with the analyses where we analyse multiple traits at once.*

There are now explicit statements about the taxonomic sampling, but there does not appear to be a mention of which species are non-homologous between the phylogenetic data sets, and what their ecological states are. This is potentially important if they are a rare state and/or positioned near the root of the tree.

With regards to sampling, we have added to the supplementary materials a diagram which will hopefully make clear the sample sizes involved in the various analyses we have done (fig. S10), as well as a list of species that are not shared between the two phylogenies (figS11). Character states for these species are included in the supplementary data.

A sentence needs to be added to the methods describing briefly what the MuSSCRat is/what it does.

We have added this sentence about MuSSCRat (line 195): ‘This allows us to test whether character states are associated with different rates of evolution beyond what would be expected from random background variation alone’

Line 146 The morphological traits are the same, it’s the dimensions of those traits that are different. This was not addressed from the previous submission

We have revised this sentence to read (line 152): ‘We used random forest classification models (38,39) to test whether differences in foraging behaviour are associated with differences in the dimensions of morphological traits.’

The ratematrix results are still largely ignored from the results section. The only sentence is “*Inferred differences between traplining, territorialist and opportunist lineages in estimated average rates of bill size and body size evolution, and their evolutionary correlation, are sensitive to the 250 phylogeny on which analyses are performed (fig. 2; fig. S1; fig. S2).*” With Fig 2 being the ratematrix results, but doesn’t describe any of the results. In particular, the lowest correlation between body size and bill size is for trapliners, which is both interesting and counter-intuitive to me not being familiar with the system. The paragraph starting on line 331 “*While we do see evidence for repeated convergent evolution of large bills relative to body size in trapliners*” I think is meant to address this, but to me the discussion points (adaptive peak shift may be old) doesn’t really address this question.

The results of ratematrix analyses are described in the paragraph beginning line 302. Because we have performed many analyses, a detailed listing of which pairwise comparisons are significantly different would add excessively to the word count and not help with understanding of the text. All of the information on pairwise comparisons is conveyed by the figures (fig. 2, fig. S1, figS2) and we now make this clear in the text. These figures express in a compact way which pairwise comparisons in which sensitivity analysis have significant overlaps in posterior distributions- indicated by colour in the rectangular panels. The key conclusion in that paragraph is ‘Given these facts and the significant overlap in posterior distributions for parameters, we conclude that there is no definitive evidence for differences in the average rate of bill size and body size evolution, or the evolutionary correlation between them, for trapliners, territorialists and opportunists.’ We hold back on drawing any conclusions about differences between groups because our sensitivity analyses show that this is not supported by a robust foundation of evidence and there are no differences that consistently stand out across different analyses. We have tried to clarify what the BayesTraits analyses are for in that

paragraph (line 274): ‘We used the results of the BayesTraits analyses of variable rates of trait evolution to identify possible causes of the inconsistent results across trees.’

Line 69: This prediction has been made before, I would say the difference here is that the authors are testing the hypothesis with a much more comprehensive sampling strategy

Yes, we agree that we are not the first to make this prediction. We have rephrased the sentence to be clear that we are addressing a prediction (rather than making a completely novel prediction) and have added citations to previous work that make similar predictions (line 74).

Line 84: sentence ending “niche more unlikely” needs supporting citations, there is a rich body of work on this topic

We have added the following sentence to bolster the point (line 84): ‘The evolution of an increasingly specialised morphology in hummingbirds to exploit specific sets of flowers may make subsequent reversal to a generalist niche more unlikely. While an adaptive ramp may be available for hummingbirds to become increasingly morphologically specialised through gradual coevolution with flowers, reversals may be hindered by an absence of a gradual adaptive ramp in the reverse direction due to competition with short-billed hummingbird species for short-tubed flowers’

We have added refs. 24 and 25

Line 86: This sentence needs supporting citations as to why more specialized species are more likely to go extinct

We have added two citations for this point and the previous one (line 92): refs 24 and 25

Revised Reply to Reviewers’ Comments

Below we expand on our previous responses to the first round of reviewers comments.

Referee 1

Line 46. The second and third sentences of the abstract could be rephrased and merged. Also, I think a sentence saying why to be a trapliner requires (or is thought to require) a specialized morphology.

This is a good suggestion. We have therefore edited the abstract to explain, in brief, why becoming a trapliner is thought to require morphological specialisation. While we cannot go into great detail in the abstract, we build on the argument more fully in the Introduction (line 40): ‘The evolution of traplining in hummingbirds is thought to entail morphological specialisation through the reciprocal coevolution of longer bills with the long-tubed flowers of widely dispersed plant species’.

Line 68. Maybe it is worth to mention that “widely dispersed flowers” refers to flowers of a single species or a small group of species.

We agree and have clarified that it is usually widely dispersed flowers of a guild of species rather than single species (line 65): ‘Through coevolution with the flowers of the various species they pollinate, trapliners may therefore become more morphologically and ecologically specialised than their non-traplining counterparts.’

Line 78. Is the bill length the only morphological feature that trapliners evolve? It is worth mentioning briefly the ones you pointed in line 92.

While multiple traits could be associated with traplining for reasons unknown to us, the literature (now cited in the text as references 18 and 19) suggests that that bill and wing are the two likeliest contenders for which a plausible adaptive story exists (line 72): ‘Specifically, we predicted that through coevolution with the flowers they pollinate trapliners should evolve relatively large bills for their body size and a higher wing surface area relative to body size.’

Line 108. I would like to see the beginning of this paragraph slightly modified. The paragraph would read better if the authors mentioned where the concept of evolutionary dead end comes from and what are the criteria (according to classic literature) that an evolutionary dead end should meet to be called as such. If irreversible transitions and lower rates of diversification are the only two points, please make it clearer. So, I would like to see a clear separation between the concept of evolutionary dead end, and the predictions the authors make for traplining.

Line 82. We thank the referee for an excellent suggestion. We have rewritten the paragraph so that it introduces the general concept of ‘evolutionary dead ends’ first and then goes on to discuss specific predictions for hummingbirds: ‘Ecological specialisation on a subset of resources could be hypothesised to be an ‘evolutionary dead end’ in the sense that evolutionary reversals back to a generalised ecology are rare and specialised species rarely give rise to new specialised species (22,23)...’

Line 118. Please add a couple of sentences explaining what the goals of this study are.

Line 95: We have added this sentence: ‘Here we test these ideas on hummingbirds using phylogenetic comparative methods to characterise diversification and rates of morphological evolution in relation to evolutionary transitions in foraging ecology.’

Line 116. The sentence about specialization increasing the chances of speciation feels out of place in here, it is a bit confusing how to reconcile this with the evolutionary dead end concept which is described above.

The statement in question is saying that the assumptions of the evolutionary dead end hypothesis may not always be met and so contrary outcomes may be possible depending on how the evolutionary dynamics of specialised species play out. A possible reason for this specific to hummingbirds is given line 93: ‘On the other hand, clades of trapliners may be able to diversify in the specific flowers on which they feed, supporting high rates of diversification, the opposite of what the evolutionary dead ends hypothesis would predict’

Line 136. How many landmarks?

*We have added the number of landmarks (4 + 3*25 semilandmarks) to this manuscript. The landmarking was done as part of a broader project on quantifying bird beak shape. We refer the reader to Cooney et al. 2017 for a more detailed explanation of the landmarking and data sources.*

Line 152. I understand sex-specific behaviour is not available for all species. For the ones with available information, what percentage of species show difference in sex-specific behaviour? I mean, how rare is that male does something different from the female?

Line 126: This is a great suggestion - it would be fascinating to examine sex differences in behaviour, unfortunately the data are very scarce. We have now added to the manuscript what little information we know from our own literature search about the sex-specific dimorphism in foraging ecology of hummingbirds: ‘There are 12 species in our assembled dataset where foraging behaviour is described for both sexes, and there is sexual dimorphism in six of these’.

Line 174. I am fully aware of the complications that come with making classifications. I understand that decisions have to be made in order to carry on. What if those facultatively territorial species were assigned to territorial category instead of opportunistic? I think the study will benefit of performing some sensitivity analysis where all analyses are repeated but using facultatively territorials as opportunistic. If general conclusions do not change, then sensitivity analysis could go to supplementary material.

This is an excellent suggestion. We have now performed sensitivity analyses where all opportunists and territorialists are simply classed as 'non-trapliners'. We have done so for the ratematrix analyses and secsse (Fig S2, Table S9). Our conclusions remain qualitatively unchanged.

Line 176. To this point, the authors have not described any analysis, so it seems out of place to clarify that analyses were repeated using two different phylogenetic reconstructions. Perhaps is better to describe the tree from reference 35 (and briefly mention that there is another tree around) and in a later stage in methods to say that the analysis was repeated using the one from ref 36 (and can be found in supplementary material).

Line 142. We have changed the order of sentences in this paragraph. 'We used two alternative published phylogenies in our comparative analyses'

Line 186. Ecological data means the foraging behaviour?

Yes, we have changed most mentions of foraging ecology to foraging behaviour to avoid confusion.

Line 196. Could you please briefly describe what random forest classification model are? Including the advantages of such a method.

Line 154. We have added the following line to the text" ' Random forest models use sets of decision trees to classify items according to multiple variables and have an advantage in accounting for potentially complex multi-dimensional relationships between predictor and response variables'

Line 202. It does not seem clear to me how you could perform a PCA using only two variables: bill centroid and body mass.

PCAs are usually performed for dimensionality reduction purposes, but there is nothing wrong in principle with performing a PCA where there are only two axes in the original data. The PCA just reorients the axes so that you have one describing the trend through the data and one describing orthogonal variation. The reason for doing a PCA here is to more closely satisfy the assumption of random forests that predictor variables split the data into groups with different compositions. Allometry seems to be a key discriminant of hummingbirds that trapline and those that don't.

Line 212. I think more details on the Brownian motion simulation are necessary to fully understand this part of methods.

Line 167. We have added further detail about the Brownian motion simulations: 'To generate a null expectation of classification accuracy, we simulated the evolution of traits randomly under Brownian motion 1000 times using the fastBM function of the R package phytools (Revell, 2012) on rate-scaled trees inferred using BayesTraits v3 with default settings (40) (<http://www.evolution.rdg.ac.uk/>; see below) and used these randomly simulated traits as predictors of foraging behaviour. We generated simulated data for bill centroid size and body mass independently.'

Line 252. Could you please clarify what that 20 means and how it was chosen?

Line 200. It was based on inspecting the tree and making a judgement about roughly how many transitions there appear to have been. 'All priors and settings were the defaults except for the prior on the number of expected number of transitions which we set to 20 based on a prior judgement of how many transitions in foraging ecology there appear to have been in the tree.'

Line 254. Should it be morphological evolution instead of just evolution?

We agree and have changed 'evolution' to 'morphological evolution' in the text where needed.

Line 328. From this first part of results, I can see that traplining strategy is not only present in hermit hummingbirds but also in other clades e.g., inca clade. However, it is not clear in the text whether inca clade is a sister clade to hermits. I wonder whether authors can provide a more convincing argument about evolutionary convergence of this traplining. From the top of my head I can think of sort of phylogenetic signal tests, but I guess there are more proper and robust ways to test this.

All of the major traplining clades we mention are not sister to each other. These instances of apparent convergence can be seen in Figure 3. We now refer the reader to figure 3 (line 329) to see where there are likely to be independent origins of traplining.

Figure 2. This figure would be easier to read if the panel in yellow is shown along with its axes

We have updated the figure legend for clarity The upper left panel is for rates of body size evolution, the bottom right for rates of bill size evolution and the top right for their evolutionary correlation

Line 412. I would like to see a more elaborated explanation on “reversals to be incomplete”.

Line 329. We have now expanded on this point: 'Multiple transitions from territorialism to traplining have taken place in the course of hummingbird evolution, while reversals from traplining to territorialism are rare or incomplete in the sense that trapliners only ever become facultatively territorial opportunists.'

Line 479. I think the authors meant “it does not lead”.

Line 384. We have changed it to 'it does not lead'

Figure 3. The figure will be more informative if there is a visual delimitation of groups (i.e., emeralds, hermits, etc).

Figure 3. We have made a physical delimitation of groups by means of arcs.

Referee 2

It appears that the supplementary material was not provided with the submission, preventing evaluation of a large portion of the results. This must be addressed before proceeding.

It is regrettable that the supplementary materials were not available to the reviewer. The other reviewers were able to access the supplementary materials . We suggest that the reviewer contact the editors for assistance.

There must be up-front statements of what the taxonomic sampling is. This is in two places 1) at the start of the data section in the methods, and 2) between the two phylogenetic hypotheses

used. Reading the methods, I assumed that taxonomic sampling was equivalent between the phylogenetic hypotheses as there was no mention of sampling; however, in the results it becomes apparent that it is not equal. Please provide raw numbers of species, what was different etc.

We agree with this important point and have added further information on number of species sampled:

Line 104: 'We took 3D scans of the entire bill and linear measurements of bill length, bill width, bill depth, wing length and tail length from one male museum specimen for each of 289 species at the Ornithological Collection of the Natural History Museum in Tring (UK).'

Line 118: 'We were able to classify 238 hummingbird species (~80% of all genera) as either trapliners (70), territorial (104) or opportunists (64).'

Line 142: 'Trees from ref. (35) (available from www.birdtree.org) are based on genetic sequence data plus taxonomic imputation for 299 species while trees from ref. (36) (available from www.tree.opentreeoflife.org) are based on genetic data only for 291 species.'

There does not appear to be any mention of the ratematrix results the corresponding section (Does the evolution of traplining entail higher rates of evolution and weaker evolutionary correlation between bill size and body size?). The paragraph talks about the BayesTraits analysis, with some mention of correlations, but no specific results. Figure 2 is only BayesTraits results.

We agree that the results of ratematrix analyses are not prominent in the text and that it was not clear when we are referring to the Bayestraits analyses and when we are discussing ratematrix. For clarity, we now state explicitly that we used the Bayestraits analysis to understand why the ratematrix analyses differed between trees. We believe that this should remove any confusion. The ratematrix analyses are presented in figure 2, figure S1 and figure S2. The overlap in posterior distributions is discussed as a means of identifying significantly distinct distributions for parameters, and the paragraph (lines 272-292) explains that whether there are any significant differences is highly sensitive to which phylogeny and which species are included. The Bayestraits analyses are brought in to try to identify which clades are driving these inconsistent results. Bayestraits results are presented in figure S3.

I feel there are significant gaps in the paragraph on assessing morphological convergence. First, was a phylogenetic PCA or non-phylogenetic PCA performed? Given the potential for phylogenetic signal in traits (with largely traplining clades) this step could potentially affect the results in a big way. Then how much variance was explained by each PC axis? Second, what software/program/packages were used for the random forest analysis? How was the initial tuning performed? How were the hyperparameters tuned? There is a lot of information that needs to be clarified here. Third, if a PCA is performed on only two variables (bill centroid size and body mass) is this not just a bivariate regression? It is unclear why a PCA would be needed when two variables are being analyzed.

We agree and have addressed this in detail in our response to reviewer 2's second review above.

There needs to be greater exploration of how different models were tested for analyses of character evolution, and subsequently implemented for ancestral state reconstructions. In the methods, the authors outline two models fit using fitDiscrete; first, one where all transition rates are estimated, and one where transitions from traplining to territorialism and from opportunism to traplining were 0. This is a bit worrying because this reads as, after fitting one model, the authors then fit a more extreme version of that model and compared the relative fit.

That would not seem to be a meaningful comparison. Are there no other plausible models that could be tested? Second, more models may have been fit, but the supplementary material does not appear to have been provided, meaning table S5 cannot be examined. Also, that the delta AIC was exactly 4 is surprising given the use of 4 as a cutoff value for differentiating models; I would like to know what the non-rounded number is here. Finally, the authors state that transitioning from traplining to territorial is irreversible (line 373), but does occur in their tree (in the emeralds). In the discussion the authors mention that this transition is incomplete at best, but I think the text could easily be made consistent.

The reduced transition rate model is a nested case of the full model where certain transition rates are fixed at zero. This is conceptually equivalent to testing whether certain transition rates are significantly different from zero, except that it's done in a model simplification framework. Following the advice of another reviewer we have tried different numbers of unique transition rates (2,4,6) in the SecSSE portion of our analyses. These analyses complement the fitDiscrete analyses. We find that in the best-fitting SecSSE model there are 4 transition rates consistent with those that we found with fitDiscrete. Line 219: 'We therefore fitted a reduced model in which these transition rates were fixed to zero and compared the full and reduced models using the Akaike Information Criterion (AIC). We used this to test for irreversibility in transitions.' We have updated table S6 with likelihoods and AIC values to nearest two decimal places. The delta AIC between models is 4.00 correct to two decimal places.

In the results paragraph starting line 361 -- doesn't the analysis performed test for a difference between groups, not pairwise, so there is a significant difference between trapliners+opportunists with territorial, not necessarily that trapliners are greater than both other groups. That is certainly how it appears looking at Figure 1.

Line 259 We have clarified this and added this to the manuscript: 'Between trapliners and opportunists, it is trapliners that tend to have the largest extremes of relative bill size, although the predicted phylogenetic regressions are very close between the two groups. It could be hypothesised that opportunists are therefore ecologically adapted to traplining since they are at least facultatively trapliners. This is not inconsistent with our hypotheses. Trapliners, and opportunists, tend to have relatively larger bills for their body size than territorialists (fig. 1).

Line 69 – it might be worth setting up some background as to why traplining is considered specialization, e.g. bill shape

We thank the reviewer for this suggestion and have added the following (Line 57: 'Trapliners are pollinators that visit widely dispersed flowers along circuitous foraging routes. The evolution of traplining in hummingbirds is thought to entail morphological specialisation through the reciprocal coevolution of longer bills with the long-tubed flowers of widely dispersed plant species.'

Line 73: The sentence starting "Widely dispersed..." doesn't follow logically from previous, suggest editing for clarity. Could maybe even flip sentence ordering with the one before which would take care of the above comment mostly.

Line 58: We have changed the sentence order to address this comment but would be happy to further edit if necessary.

I would recommend moving the line 172—174 on the numbers of species in each ecological state (possibly with modification) to the start of the corresponding paragraph as knowing how many states there are would make reading the paragraph much easier.

We have brought this to the beginning of the paragraph: ‘We were able to classify 238 hummingbird species (~80% of all genera) as either trapliners (70), territorial (104) or opportunists (64)’

I think there could be more of a setup for the counter argument to the “evolutionary dead-end” hypothesis in the introduction, especially given the results of the study. For example, why can't selection towards a more "generalist" bill shape be equally strong given the costs noted by the authors about becoming a trapliner. It is not quite clear to me why the selection has to be unidirectional – is there a event horizon (for want of a better term) where selection for shorter bills cannot happen?

Line 86. We have rewritten the last paragraph of the Introduction to present opposing hypotheses on the consequences of specialisation.: ‘While an adaptive ramp may be available for hummingbirds to become increasingly morphologically specialised through gradual coevolution with flowers, reversals may be hindered by an absence of a gradual adaptive ramp in the reverse direction due to competition with short-billed hummingbird species for short-tubed flowers’

Line 145: nix “and from” and the end of the line

deleted ‘and from’

Line 198: sentence ending “different sets of morphological traits” -> the traits are the same, it's the measurements/landmarks of those traits that differ between ecological states

Line 153. We agree that the wording was unclear and have edited it to read: ‘We used random forest classification models (38,39) to test whether differences in foraging behaviour are associated with differences in morphological traits’

Referee 3

State-dependent analyses are very sensitive to the way the trait data is handled. I find it problematic to treat “opportunist” as a distinct character state. As it was mentioned several times throughout the manuscript, this is an intermediate form of foraging that exhibits both traplining and territorial behaviour with varying degrees. I realize it is difficult to formulate this behaviour as a continuous trait, one option would be to exclude “opportunist” taxa from the analysis and perform the analysis only with the “traplining” and “territorial” taxa. Alternatively, the presence and absence of territorial (or traplining) behaviour could be coded as 1 and 0 respectively to examine the effect of one state at a time.

We agree that such analyses can be sensitive to character coding. This issue was also raised by other reviewers. We have now performed sensitivity analyses where all opportunists and territorialists are simply classed as ‘non-trapliners’. We have done so for the ratematrix analyses and secsse. Our conclusions remain qualitatively unchanged (fig. S2, table S9).

Figure 1: The data is not provided to see which data point refers to which species. As it is, there are some interesting outliers that the readers might be interested in knowing more about. OK I see the data file now. It should be cited in the text, so the readers can easily find it.

Figure 1: The data will be made available with a link upon publication. There is a placeholder link where the data can be accessed.

Figure 2: The coloured rectangles are extremely difficult to interpret and the figure legend does not help with this either. This part of the figure could be converted to a table instead. Are those p values in the middle of each rectangle? (Same comment for Figure S2)

Figure 2: We acknowledge that the rectangles are a somewhat unusual graphical device, but we feel that it is the best way to present the information given the complexity of the results. Pairwise overlaps in posterior distributions are not p-values- but similar to p-values an overlap of less than 5% is usually considered a 'significant' difference.

Figure 3: What does the arrow with a red hummingbird silhouette refer to? The clade boundaries should be more clear; we cannot see which species are included within the emeralds, coquettes...etc.. Most of the species names are not legible on the tree. The legend needs more description for the ancestral reconstruction (e.g., what do the coloured pies represent? What do the asterisks mean?) Some of these might seem obvious, but not for everyone.

Figure 3: In the legend to figure 3 we have named the species that the arrow points to. We have also added descriptions of what the pies and asterisks mean. We have added solid arcs to physically delimit clades of interest. The species names are there in case anyone wants to download the pdf and zoom in on a clade of interest to see which species are there.

Figure S2: Both trees have “outlier” taxa with long branches. Are those the ones that were pruned. Some clarification would be nice. It would also be useful to see the topological differences between the two trees (e.g., by highlighting the differently placed clades).

Figure S3: We do not know exactly how we would show the topological differences between the trees. We have added the trees as nexus files so that the reviewer can examine the topological differences as they wish. Figure S3 legend: 'The long branches are ones which were pruned in sensitivity analyses.'

Figure S5: This should be a table, not a figure (it says table in the main text; same for Figure/Table S6). What constraint is used in the second model? It should be described here so that readers don't have to go back to the main text. In the main text, it says AICc was used, but the table here says AIC. What is n and which one is used?

Figure S5: We have relabelled it as a table. AICc was a typo which we've now corrected. It should be AIC.

Referee 4

Methods: The first sentence lacks the information about the exact number of species that have been measured. Please also indicate the number of species included in each of the 2 phylogenetic tree. This way one could get a clear picture of the overlap between the morphological, ecological and phylogenetic dataset. Furthermore, Information about sampling in such kind of study is necessary given that the statistical power of PCM is highly dependent on sampling proportion.

Methods: We have added sampling information to the Methods section. See reply to reviewer 2 for details

L196. A brief explanation of random forest classification model would be useful.

Line 154: We have added a brief description as suggested: 'Random forest models use sets of decision trees to classify items according to multiple variables and have an advantage in accounting for potentially complex multi-dimensional relationships between predictor and response variables.'

L.212. by rate-scaled trees do you mean trees with branch length proportional to the rate of evolution of the 2 morphological traits? If so, what is the rationale for using such trees when simulating the evolution of these 2 traits?

Yes, rate scaled trees inferred using BayesTraits have branches scaled to the proportional rate of evolution for each trait. This is not strictly necessary, but the rate variable model is closer to the true evolutionary process that has produced the present trait distribution. This leads to a more biologically realistic null model.

L.216-218. Could you please specify if you used trait values or PC scores for the pgl's?

We used the original trait values. We have clarified this in the text. Line 174

L371. Had a better fit rather than was more parsimonious seems more correct

Line 298: We agree and have revised the text to read: 'Reversals from traplining to territorialism are rarer still. We found that a model in which the transition rate from traplining to territorialism was fixed to zero had a better fit than a model in which all transition rate parameters were free to vary.;

Fig.1 Please mention in the caption that it shows the result of PGLS regressions.

Figure 1: We have added PGLS to the legend

Fig.3. Some of the tips' names are readable but most of them are completely distorted.

Figure 3: We apologise for this issue. However, we have not been able to replicate the problem so are unsure how to address it.

Appendix D

Response to Referees

Referee: 2

Comments to the Author(s).

This is my third reading of this manuscript, and I am extremely pleased to see the detail with which the authors have addressed this latest revision. It enabled me to evaluate the strength of their results in the context of pretty common concerns. That some of the key results don't change while accommodating these concerns only serves to strengthen their conclusions, a more desirable outcome than disregarding issues raised by reviewers. I still think this is a really interesting paper, and I think it will ultimately be a useful contribution to the field. I don't have any major methodological concerns, most of my in-line comments I feel are subjective with the exception of two, which I highlight below.

Minor comments

Line 277: would move the sentence "the results of all pairwise..." inside the parentheses of the supplementary figures

We have moved the sentence inside the parentheses.

Line 338: Sentence starting "This could be explained" took me a couple of reads, should the sentence be "tripliners also become morphologically specialized to a niche...", perhaps revisit this sentence for clarity.

We have rephrased the sentence. These two changes can be seen in the tracked changes document below.